



# MONODEUTERATED METHANE:
# AN ISOTOPIC PROBE TO MEASURE BIOLOGICAL METHANE METABOLISM
# RATES AND TRACK CATABOLIC EXCHANGE REACTIONS

Jeffrey J. Marlow[1][*][#], Joshua A. Steele[1][^], Wiebke Ziebis[2], Silvan Scheller[1],
David Case[1], Victoria J. Orphan[1]

[1] Division of Geological and Planetary Sciences, California Institute of Technology, Pasadena, CA, 91125 USA
[2] Department of Biological Science, University of Southern California, Los Angeles, CA, 90089 USA
* Current address: Dept. of Organismic and Evolutionary Biology, Harvard University, Cambridge, MA 02138 USA
^ Current address: Southern California Coastal Water Research Project, Costa Mesa, CA 92626 USA
# Correspondence email: marlow@fas.harvard.edu



**Abstract**

7       Biological methane oxidation is a globally relevant process that mediates the flux of an

important greenhouse gas through both aerobic and anaerobic metabolic pathways. However,
measuring the rates of these metabolisms presents many obstacles, from logistical barriers to
regulatory hurdles and poor precision. Here we present a new approach for measuring rates of
microbial methane metabolism that is non-toxic, rapid, and relatively high throughput, alleviating
some of the current methodological challenges. Specifically, we tested the potential for using
monodeuterated methane ($CH_3D$) as a metabolic substrate for measuring the rate of methane
activation by quantifying the change in the aqueous D/H ratio over time using a water isotope
analyzer. This method represents a non-toxic, comparatively rapid and straightforward approach
that is complementary to existing radio ($^{14}C$)- and stable ($^{13}C$) carbon isotopic methods; by probing
hydrogen atom dynamics, it offers an additional dimension through which to examine the rates and
pathways of methane metabolism. We provide direct comparisons between the $CH_3D$ procedure
and the well-established $^{14}CH_4$ radiotracer approach for several methanotrophic systems, including
type I and type II aerobic methanotroph cultures, and methane seep sediment and carbonate rocks
under anoxic and oxic incubation conditions. We also employ this method to investigate the role of
pressure on methane oxidation rates in anoxic seep sediment, revealing an 80% increase at the
equivalent of ~900 m water depth (40 MPa).
The monodeuterated methane approach offers a procedurally straightforward, reliable
method that advances three specific aims: 1) the direct comparison of methane oxidation rates
between different experimental treatments of the same inoculum; 2) the determination of an
absolute scaling factor using paired $CH_3D$ and $^{14}C$-radiocarbon procedures for new systems of
interest; and 3) a continued evaluation of C- and H-atom tracking through methanotrophic



metabolisms, with specific foci on enzyme reversibility and anabolic/catabolic branch points. The
procedural advantages, consistency, and novel research questions enabled by the monodeuterated
methane method should prove useful in a wide range of culture-based and environmental microbial
systems to further elucidate methane metabolism dynamics.
**1. Introduction**
Methane-consuming microbial processes represent an important component of
biogeochemical cycles in natural freshwater and marine environments, as well as in human-
impacted systems. In terrestrial soils, methane production in rice fields, anoxic wetlands, and
thawing permafrost supports methanotrophic communities (Holzapfel-Pschorn et al., 1985;
Mackelprang et al., 2011). In marine settings, an estimated 85 Tg of methane per year, derived
from biogenic and thermogenic sources, enters the subseafloor, the vast majority of which is
anaerobically consumed in anoxic sediments (Reeburgh, 2007). Much of what remains is taken up
in microoxic or oxic zones of the sediment or water column by aerobic methanotrophic
microorganisms (Valentine et al., 2001). Methanotrophy is also of interest in a range of human-
impacted contexts, including wastewater treatment plants (Ho et al., 2013), landfills (Scheutz et al.,
2009), and oil spills (Crespo-Medina et al., 2014).
In addition to the climatic and economic implications of the methanotrophic process, its
biochemical intricacies have stimulated many investigations. The anaerobic oxidation of methane
(AOM) has proven particularly enigmatic, often involving a mutualistic relationship between
anaerobic methanotrophic (ANME) archaea and sulfate reducing bacteria (SRB; Boetius et al.,
2000). A consensus on the precise nature of the mutualism remains outstanding, but the net result
of the process is typically the stoichiometric oxidation of methane coupled with sulfate reduction



(Knittel and Boetius, 2009). Alternative electron acceptors including nitrate (Haroon et al., 2013),
and nitrite (Ettwig et al., 2010) have been demonstrated, while several studies have presented
equivocal evidence for methane oxidation coupled directly to iron or manganese reduction (Beal et
al., 2009; Nauhaus et al., 2005; Sivan et al., 2014).

Methanotrophy is both a biogeochemically relevant reaction that modulates climate forcing

and a biochemical curiosity; given this dual role, there is substantial interest in measuring the rate
of the process and understanding elemental flows through metabolic pathways. AOM rate
measurements have traditionally been conducted using a handful of techniques. Numerical models
incorporating environmental sediment profiles of sulfate and methane concentrations can be used
to back-calculate methane consumption rates (Jørgensen et al., 2001). Stable isotope $^{13}CH_4$ tracers
can be used to probe longer-term rates in controlled conditions (Moran et al., 2008), but high levels
of natural $^{13}C$ in marine dissolved inorganic carbon pools complicate the measurement (Pack et al.,
2011). Gas chromatography quantification of dissolved (Girguis et al., 2003) or headspace (Carini
et al., 2003) methane concentrations has also been demonstrated as a rate measurement tool,
though low molarities make samples susceptible to exsolution if not processed quickly after
collection, a requirement that may not be achievable in field settings. Perhaps the most sensitive
approach uses radiolabeled $^{14}CH_4$ to track carbon movement into oxidized species (Alperin and
Reeburgh, 1985; Treude et al., 2003). Tritiated methane was introduced for water column aerobic
methane oxidation measurements due to its improved specific activity and the procedural
advantages of working with a water-phase product rather than gaseous products (Valentine et al.,
2001). Logistical and health and safety regulations led Pack et al. (2011) to develop an accelerator
mass spectrometry detection method that requires $10^3$-$10^5$ less radiolabel than previous $^{14}C$ and $^3H$
approaches, though the analytical procedure remains labor intensive.



Despite the range of procedural options, methanotrophy rate measurements remain

cumbersome, and the demonstration of a precise, safe, and easily enacted approach would be a

welcome contribution for a diverse field of researchers. Nearly all of the aforementioned

approaches are carbon-based; a hydrogen-based tracer offers an additional dimension to

investigations of methane biochemical dynamics. Here we introduce a novel method for

biologically mediated methanotrophy rate measurement that utilizes monodeuterated methane

($CH_3D$) as a substrate and measures the D/H ratio of the aqueous solution.

We demonstrate, through methanotrophic cell cultures and microcosm incubations of

seafloor sediment and carbonate rock fragments, that aqueous D/H values are consistently

proportional to $^{14}C$-based rate measurements for given laboratory treatments tested in this study.

The resulting ratios, when viewed in the context of partial versus complete methane oxidation,

represent a new tool with which to examine the reversibility and catabolic / anabolic partitioning of

methanotrophic metabolisms. As a rate measurement protocol, this approach offers several

advantages over current techniques: it does not require the logistical, safety, and administrative

hurdles associated with radiotracers such as $^{14}CH_4$ and $^3H$-$CH_4$, it is less susceptible to analyte loss

than methane headspace measurements, and compares favorably in terms of equipment cost and

portability. The monodeuterated methane protocol offers new flexibility for practitioners and

represents a useful and distinct new tool for the rate measurement of methanotrophic processes.

**2. Methods**

2.1. Experimental Set-Up

To demonstrate the precision and reproducibility of the monodeuterated methane approach,

it was tested alongside the better-established $^{14}CH_4$ radiotracer protocol. Both techniques were



applied to a) aerobic methanotrophic cultures of *Methylosinus trichosporium* and
*Methyloprofundus sedimenti*, b) oxic incubations of methane seep sediment and carbonate rocks,
and c) anoxic incubations of methane seep sediment and carbonate rocks. In addition, the
monodeuterated methane protocol was employed on its own to demonstrate the relative effect of
heightened, environmentally relevant pressure on methane consumption rates in anoxic seep
sediment samples. Monodeuterated methane for all samples was 98% pure $CH_3D$ obtained from
Sigma-Aldrich ($247 / L). For a representation of all experiments conducted in this study, see
Table 1.
*2.1.1. Aerobic Methanotroph Cultures*
Cultures of *Methylosinus trichosporium* strain OB3b (Whittenbury et al., 1970) were
grown using Nitrate Mineral Salts (NMS) medium at 30 °C. The newly characterized
*Methyloprofundus sedimenti* strain WF1 was grown in a modified NMS medium at 25 °C
(Tavormina et al., 2015). In both cases, shaking cultures were grown up from stock in sealed 25
mL test tubes that contained 5 mL media and 50:50 air:methane by volume. After several
successful transfers, experiments were initiated by passaging 0.94 mL of exponential phase
inoculum into 8.5 mL media, for each of ten different experimental conditions, each prepared in
triplicate (see Table S1). These conditions tested the $CH_3D$ approach against killed, cell-free,
oxygen-free, and $CH_3D$-free controls; parallel incubations incorporated $^{14}CH_4$ to allow for three
time points of destructive sampling, and killed and radiolabel-free controls.
Samples for D/H analysis were taken at seven time points – most concentrated around
anticipated exponential growth phases – throughout 140-hour (*M. trichosporium*) and 476-hour
(*M. sedimenti*) experiments. Samples for radiolabel processing were taken at 46, 102, and 166.5



hours for *M. trichosporium* cultures and 102, 166.5, and 432 hours for the slower-growing *M.*
*sedimenti* cultures.
*2.1.2. Environmental Samples: Methane Seep Sediments and Carbonates*

123          Samples recovered from the Hydrate Ridge methane seep system were used to

comparatively examine the novel monodeuterated methane ($CH_3D$) approach alongside the $^{14}CH_4$
protocol with environmental samples. Hydrate Ridge, Oregon, is located along a convergent
tectonic margin and is well established as a site of methane seepage and sediment-based AOM
(e.g., Suess et al., 1999; Treude et al., 2003; Tryon et al., 2002). Methane concentrations within the
most active seep sediments reach several mM, and have been measured and modeled at values up
to 70 mM (Boetius and Suess, 2004) and 50 mM (Tryon et al., 2002) respectively.

130          Samples used for methanotrophic rate experiments are specified in Table 1. All samples

received a unique four-digit serial number. The "active" designation refers to sites where methane
seepage was manifested by seafloor ecosystems known to be fueled by subsurface methane (e.g.
clam beds and microbial mats) or methane bubble ebullition. The term "low activity" references
sampling sites that did not exhibit any clear signs of contemporary methane seepage or
chemosynthetic communities, though a small amount of methane supply and methanotrophic
potential cannot be ruled out as subsurface advective flow can shift with time (Gieskes et al., 2005;
Marlow et al., 2014; Tryon et al., 2002). The presence of carbonate pavements, coupled to depleted
$\delta^{13}C_{carbonate}$ values suggest that they formed during "active" periods of seepage, consistent with
previous descriptions (Naehr et al., 2007; Peckmann and Thiel, 2004). Sample types are
abbreviated by the A.Sed (active sediment), A.Carb (active carbonate), L.Sed (low-activity
sediment), and L.Carb (low-activity carbonate) designations. Seven samples were analyzed to
examine a range of physical substrate type (sediment vs. carbonate rock) and seepage



environments (active and low-activity): A.Sed-5128, A.Carb-5305, A.Carb-5152, L.Sed-5043,
L.Carb-5028, and sterilized control aliquots of A.Sed-5128 and A.Carb-5305. Carbonate samples
include both porous materials with macroscale vugs and pore spaces, as well as massive lithologies
with more homogenous structure.

Samples were collected with the Deep Submergence Vehicle (DSV) *Alvin* during *Atlantis*

leg AT-16-68 in September 2010 and the Remotely Operated Vehicle (ROV) *Jason* II during
*Atlantis* leg AT-18-10 in September 2011. Shipboard, push cores and bottom water-submerged
carbonates were immediately transferred to a 4 °C walk-in cold room and processed within several
hours. To prepare material for future experimentation, compacted sediment and carbonate rocks
were stored in anoxic, Ar-flushed mylar bags at 4 °C until use. In advance of experimental set-up,
sediment and carbonate samples were prepared under anoxic conditions using 0.22 μm-filtered,
anoxic $N_2$-sparged Hydrate Ridge bottom water (at a 1:2 sediment/carbonate:bottom water ratio by
volume) and maintained under a $2 \times 10^5$ Pa $CH_4$ headspace for one month.

To set up the experimental incubations, 10 mL physical substrate (compressed sediment or

carbonate rock) and 20 mL filtered Hydrate Ridge bottom water were placed in 60-mL glass
bottles (SVG-50 gaschro vials, Nichiden Riku Glass Co, Kobe, Japan). In all experiments
involving carbonates, interior portions (> 5 cm from the rock surface) were used in order to ensure
that properties exhibited were representative of bulk rock material and not a reflection of surface-
based adherent cells or entrained material. Carbonate rock samples were fragmented in order to fit
through the 28-mm diameter bottle opening; pieces were kept as large as possible to minimize the
increase in surface area-to-volume ratio and maintain conditions as representative of the *in situ*
environment as possible. All bottles were sealed with butyl stoppers; following several minutes of
flushing with $N_2$ (g), the headspace was replaced with methane, and an additional 30 mL of gas



was injected into the 30 mL headspace to generate the desired headspace composition at an
absolute pressure of approximately $2 \times 10^5$ Pa. Anoxic incubation headspace was 100% methane;
oxic incubation headspace was 30 mL methane, 20 mL $N_2$, and 10 mL $O_2$. All incubation set-up
prior to gas flushing and headspace injection took place in an anaerobic chamber. Triplicate
samples, including killed controls, were prepared for all sample types. Measurements were taken
for both D/H and $^{14}C$ analysis at 1.9 and 4 days for oxic incubations, and 3 and 8 days for anoxic
incubations. Anoxic active methane seep sediment (A.Sed-5128) incubations were used for
isotopic analysis of the remaining methane (set up in triplicate, with 60 mL $CH_3D$ initial
headspace) as well as empirical resolution studies sampled between days 20 and 22.
*2.1.3. Pressurized samples*
In order to probe the effect of pressure on anaerobic methanotrophic rates, a set of
experiments was established, using the monodeuterated methane technique to determine relative
rate differences. Active sediment from Hydrate Ridge (A.Sed-3450) was collected, processed
shipboard, and prepared for experimentation as described above. To set up the incubations, eight
100 mL mylar bags were prepared with the components shown in Table S2: identical sets of four
compositionally distinct samples were established such that each could be subjected to low and
high pressure. Prior to gas addition, each bag was flushed for 5 minutes with Ar.
Once the incubations were prepared, they were transported to the laboratory of Dr. William
Berelson at the University of Southern California and inserted into a stainless steel, custom-built
pressure chamber with 3-cm thick walls and pressure valves rated to 40 MPa. The chamber was
placed in a walk-in cold room (4 °C) on site, and hydraulic fluid was pumped into the sealed
chamber using a Star Hydraulics P1A-250 hand pump. The pressure was maintained at 9.0 MPa
(equivalent to ~900 m water depth) during the course of the 38-day experiment, with daily





adjustments to account for thermal compression effects. At the conclusion of the experiment, mylar
bags were removed from the chamber and checked for leaks (none were observed) and sampled for
D/H ratio measurement.
2.2. Analytical Procedures
*2.2.1. CH₃D Rate Measurements*

*2.2.1. $CH_3D$ Rate Measurements*
At designated sampling times, ~1 mL of medium / water was collected from cultures or
incubations in an anaerobic chamber with a sterile syringe. The liquid was then pushed through a
0.22 μm Durapore filter (EMD Millipore, Temecula, CA) and into a 1-mL GC vial. A LGR DLT-
100 liquid water isotope analyzer (Los Gatos Research, Mountain View, CA) was used to
determine the D/H ratio of each sample, with an injection volume of 700 nL at 1000 nL/s, four
intra-injection flush strokes, and a flush time of 60 s between injections. Four rounds of ten
injections per sample were performed in order to avoid memory effects; only the latter five
injections were used in subsequent calculations. Sample runs were limited to ~250 injections in
order to minimize salt precipitation, and each analysis included an appropriate blank (i.e.,
autoclaved media for the cultures, or filter sterilized bottom water used during incubation set-up in
the case of sediment and carbonate incubations) and two standards of known isotopic ratios (Deep
Blue: $\delta D = 0.5‰$, and CIT: $\delta D = -73.4‰$). Data was removed if instrumental temperature or
pressure parameters were flagged as sub-optimal (0.76% of all analyses).
To calculate methane consumption rates, the number of deuterium atoms in the culture /
incubation was calculated using the experiment's overall water volume and the adjusted D/H
values (averaging across the latter five injections of the four distinct injection rounds). This value
was multiplied by four given the 1:3 D:H stoichiometry of the $CH_3D$ substrate to derive the
number of methane molecules consumed through initial C-X bond activation. Known D/H ratios of



the water standards were first used to generate a linear scaling factor that was applied to the
corresponding data. To minimize instrumental drift, standards were re-measured between rounds of
sample analysis (maximum of 40 injections) and new scaling factors were implemented. The
scaling factor of four was used in the context of methane activation – the initial mobilization of the
molecule through conversion to a methyl group – and is an end-member case that may not be
appropriate for all subsequent processing as hydrogen/deuterium atoms are removed or exchanged.
(Caveats and interpretation are discussed below, but consistent scaling factor implementation is the
primary requirement for reliable comparison.) The resulting proxy value was divided by the
incubation time and volume to arrive at a rate of methane consumption.
*2.2.2. $^{14}CH_4$ Rate Measurements*

Methane oxidation rates using radiolabeled methane substrate were measured as described

in detail by Treude et al. (2005) and Treude and Ziebis (2010). Radiolabeled methane ($^{14}CH_4$
dissolved in seawater, corresponding to an activity of 13 kBq for culture experiments and 52 kBq
in sediment and carbonate samples) was injected into each sample container, and samples were
incubated at the appropriate temperatures for the designated amount of time (see above). To stop
microbial activity and begin analysis, 2.5 ml of 2.5% NaOH was injected. Sample headspace was
flowed through a $Cu^{2+}$ oxide-filled 850 °C quartz tube furnace, combusting unreacted $^{14}CH_4$ to
$^{14}CO_2$. This $^{14}CO_2$ was collected in two scintillation vials (23 ml volume) pre-filled with 7 ml
phenylethylamine and 1 ml 2-methoxyethanol, to which 10 ml of scintillation cocktail (Ultima
Gold XR, PerkinElmer) was added. After a 24-hour incubation period, radioactivity from $^{14}CO_2$
was measured by scintillation counting (Beckman Coulter LS 6500 Multi-Purpose Scintillation
Counter, 10 minute analysis per sample).



$^{14}CO_2$ and $H^{14}CO_3^-$ produced during the experimental period was quantified as follows.
The entire volume of each incubation sample was transferred into a 250-ml Erlenmeyer flask along
with 1 drop of antifoam and 5 ml of 6M HCl. The flask was immediately stoppered and sealed
with two clamps and parafilm wrapping to prevent gas escape, and placed on a shaking table (60
rpm, room temperature, 24 hours). To collect $^{14}CO_2$ generated by the acidification process, a 7-ml
scintillation vial was pre-filled with 1 ml of 2.5% NaOH and 1 ml of phenylethylamine and
suspended from the rubber stopper inside the flask. After the shaking / acidification step, 5 ml of
scintillation cocktail was added, and the vial was measured by scintillation counting after 24 hours.
This method has been demonstrated to recover 98% of $^{14}CO_2$ on average (Treude et al., 2003).
Finally, sterilized control samples (#10, see Table S2) were set aside after $^{14}CH_4$ addition
for gas chromatography to determine the initial concentration of methane gas. 400 μl of headspace
was injected into a gas chromatograph (Shimadzu GC-2014), equipped with a packed stainless
steel Supelco Custom Column (50/50 mixture, 80/100 Porapak N support, 80/100 Porapak Q
column, 6 ft x 1/8 in) and a flame ionization detector. The carrier gas was helium at a flow rate of
30 ml min$^{-1}$, and the column temperature was 60 °C. Results were scaled based on comparison
with standards of known methane concentrations (10 and 100 ppm; Matheson Tri-Gas, Twinsburg,
OH). The rate of methane oxidation was determined by the equation
$$Methane\ Oxidation = \frac{^{14}CO_2 \bullet CH_4}{(^{14}CH_4 + {}^{14}CO_2) \bullet v \bullet t}$$

in which $^{14}CH_4$ is the combusted unreacted radiolabeled methane, $^{14}CO_2$ represents the quantity of
acidified oxidation product, $CH_4$ signifies the initial quantity of methane in the experiment, v is the
volume of sediment or carbonate rock, and t is the time over which the incubation was active.
*Isotopic Analysis of Methane in the Headspace*



The methane headspace was analyzed via $^1$H-NMR spectroscopy using a Varian 400 MHz
Spectrometer with a broadband auto-tune OneProbe. 300 μl of headspace was passed through
CDCl$_3$ with a fine needle to absorb the methane. $^1$H-NMR spectra were acquired at 298 K without
spinning, using a repetition rate of 10 s to ensure reliable quantification. The spectra were
simulated with the iNMR 4.1.7 software for the determination of the fractional abundances of the
$^{12}$CH$_4$, $^{12}$CH$_3$D, $^{13}$CH$_4$ and $^{13}$CH$_3$D isotopologs.

**3. Results and Discussion**
3.1. Aerobic Methanotroph Cultures
D/H ratios were acquired at eight points during the *M. trichosporium* growth curve and
seven points of the *M. sedimenti* growth curve; three measurements of $^{14}$C distributions were
acquired for each strain, targeting exponential and stationary phases (Fig. 1). The Type II
alphaproteobacterial methanotroph *M. trichosporium* exhibited methane consumption rates more
than an order of magnitude greater than those of *M. sedimenti* (gammaproteobacterial Type I
methanotroph), yet the scaling factor relating the CH$_3$D- and $^{14}$CH$_4$-derived rates was remarkably
consistent in both cases. Using data points from both CH$_3$D and $^{14}$CH$_4$ experiments taken closest to
the end of exponential growth phase (47.5 hours for *M. trichosporium*, 140 and 102 hours for *M.*
*sedimenti* CH$_3$D and $^{14}$CH$_4$ measurements, respectively) and in stationary phase (140 and 166.5
hours for *M. trichosporium* CH$_3$D and $^{14}$CH$_4$ measurements; 476 and 432 hours for *M. sedimenti*
CH$_3$D and $^{14}$CH$_4$ measurements), the ratio of methane oxidation rates derived from each approach
can be compared. This value is hereafter referred to as the "H:C tracer ratio" because the CH$_3$D
method tracks hydrogen atoms, while the $^{14}$CH$_4$ approach traces carbon atoms (see "Understanding
the H:C Tracer Ratio", below). This ratio can be used to evaluate the consistency of the





monodeuterated methane method compared with the better-established $^{14}CH_4$ approach, and as an
investigatory tool in catabolic / anabolic processing of methane.

Using averaged values of tubes #1a, #1b, and #1c for $CH_3D$ rates and the triplicate $^{14}CH_4$

tubes of the appropriate time point (#6, #7, or #8), H:C tracer ratio values were calculated and are
shown in Table 2; their consistency is a promising indicator of the utility of the monodeuterated
methane approach for ground-truthed rate measurements. By dividing rates derived from D/H
values by 1.5, a reliable estimate of full-oxidation methanotrophy can be attained.
3.2. Environmental Samples: Methanotrophy Under Oxic and Anoxic Conditions

Oxidation rates under oxic microcosm incubation conditions, derived from both $CH_3D$ and

$^{14}CH_4$ measurements, are provided for all five sample types (active sediment, low-activity
sediment, active porous carbonate, active massive carbonate, and low-activity massive carbonate)
in Fig. 2a. The corresponding values for anoxic conditions are shown in Fig. 2b; all values were
calculated from the second time point (4d for oxic conditions, 8d for anoxic conditions).

The H:C tracer ratio for the oxic incubations was 1.66 +/- 0.02 SE and 1.99 +/- 0.04 SE for

anoxic conditions (Table 2). These relatively consistent values across physical substrate type
(sediment and carbonates of varying lithology) and collection site activity level (active and low-
activity) suggest an underlying metabolic basis of these H:C tracer ratios that is unperturbed by
physicochemical factors or relative activity levels.

To determine the minimum number of activated $CH_3D$ molecules needed for analytical

detection, we assessed the length of time required to measure a differentiable D/H ratio.
Measurements were acquired at multiple time points between days 20 and 22 of a triplicate set of
A.Sed-5128 incubations. A resolvable signal of an enhanced D/H ratio was defined as data points
with non-overlapping confidence intervals, representing a 95% statistical probability that D/H





ratios were increased. Such differentiation seen at the 20-hour sampling time for two replicates and
the 26-hour sampling time for the other one (Fig. S1). Using the rate determined by the first 20
days as a baseline, this translates to a resolution of 4.5-6.2 μmol of fully oxidized methane based
on the H:C tracer ratio of 2.05 (Table 2).
3.3. Understanding the H:C Tracer Ratio

The $CH_3D$ and $^{14}CH_4$ approaches quantify distinct aspects of methanotrophy, and each

offers an important dimension in understanding methane metabolism. The $^{14}CH_4$ technique
quantifies the amount of $^{14}C$ – initially supplied as methane – that is fully oxidized and persists as
soluble species ($HCO_3^-$) or acid-labile precipitation products ($CaCO_3$). The $CH_3D$ protocol, on the
other hand, reports the extent to which methane-derived hydrogen atoms are found in the aqueous
phase. Because methane is an inert molecule, D-H exchange of monodeuterated methane with
water is negligible – an expectation borne out by the lack of significantly heightened D/H ratios in
killed control experiments (e.g., Fig. 1). Its activation thereby indicates enzymatic
functionalization, but the ultimate fate of each hydrogen during methane oxidation is unclear.

The flow of methane-derived hydrogen atoms through anaerobic and aerobic

methanotrophic metabolisms was examined in an attempt to predictively evaluate the consequence
of monodeuterated methane reactions. Previously published reports were used to compile Figure 3
(Hallam et al., 2004; Thauer, 2011; Vorholt and Thauer, 1997) and Figure 4 (Lieberman and
Rosenzweig, 2004), which trace anaerobic and aerobic methane metabolisms, respectively, with a
specific focus on hydrogen atoms. In this context, our observations of relatively consistent but
distinct H:C tracer ratios for anaerobic and aerobic methanotrophy (Table 2) likely reflect different
aspects of the two metabolic pathways. In AOM, metabolite backflux (Holler et al., 2011) may



increase the D/H ratio; in aerobic methanotrophy, biomass growth represents a substantial carbon
and hydrogen shunt.

### 3.3.1. The H:C Tracer Ratio in Anaerobic Methanotrophy

AOM is depicted via the reverse methanogenesis pathway in Fig. 3, whereby methyl-
coenzyme M reductase (Mcr) activates methane and generates methyl-CoM. A
tetrahydromethanopterin molecule supplants CoM, and subsequent carbon oxidation steps release
hydrogen atoms into the medium. Ultimately, the number of methane-derived hydrogen atoms that
enter water-exchangeable products dictates the physiological interpretation of waterborne D/H
ratios. For example, if just one methane-derived hydrogen enters an intermediate and is freely
exchangeable with water, then observed water-based deuterium must be multiplied by four (to
account for methane's hydrogen-carbon stoichiometry) and a primary isotope effect as high as 2.44
(*M. marburgensis'* Mcr's C-H vs. C-D bond-breaking preference, Scheller et al., 2013) to arrive at
the actual quantity of activated methane molecules. In this context, the experimental H:C tracer
ratio values may provide useful insight. A H:C tracer ratio of 2 for the reverse methanogenesis
pathway suggests that, for every methane molecule that is fully oxidized to $CO_2$, two hydrogen
atoms enter water-exchangeable intermediates.
However, heightened D/H ratios may occur in the absence of full carbon oxidation and
could be partially attributable to the back reaction of enzymatic processes (Scheller et al., 2010)
involving hydrogen exchange with the aqueous medium. For example, upon the activation of
methane by Mcr, HS-CoB and $CH_3$-S-CoM form, with the thiol hydrogen exchanging with water-
bound hydrogen. If the S-bound hydrogen were deuterium, then the re-formation of methane ($CH_4$)
would result in a heightened aqueous D/H ratio but no net methane consumption (Fig. 3). We
analyzed the remaining headspace for the formation of $CH_4$ from $CH_3D$ via $^1$H-NMR





spectroscopy. Over the course of 58 days in triplicate seep sediment incubations prepared with
exclusively $CH_3D$ headspace, $CH_4$ in the headspace increased from 0.33% +/- 0.02% SE to 4.48%
+/- 0.27% SE. If this demonstrated reversibility only reflects the back reaction of Mcr, then the
$CH_4$ increase must be multiplied by four to reflect the actual percentage of headspace methane that
was re-formed by Mcr; if the reversibility reflects back reaction of the entire pathway, then no
scaling factor is needed. Thus, the range of potential methane headspace percentage accounted for
by methane reformed from initial $CH_3D$ is between 4.15 – 16.6%. For clarity, these calculations
neglect isotope effects and activity by methanogens, factors that can be clarified through further
experimentation. For example, reversibility can be evaluated by a) including a $^{13}$C-dissolved
inorganic carbon (DIC) signal in the water and measuring $^{13}CH_4$, and/or b) utilizing multiply
deuterated methane as initial headspace and quantifying all possible isotopologs. Nonetheless, even
the upper bound of partially and reversibly oxidized $CH_3D$ suggests that the majority of the D/H
signal is attributable to reactions indicative of net methane consumption, if not complete oxidation.
*3.3.2. The H:C Tracer Ratio in Aerobic Methanotrophy*
In aerobic methanotrophic cultures, a H:C tracer ratio of ~1.5 was observed, suggesting that
on average, 2.67 of the four methane-derived hydrogen atoms likely enter water-exchangeable
products during the course of a full oxidation pathway. Intriguingly, this ratio was similar for both
cultured organisms despite their distinct metabolic pathways. *M. tricosporium* is a type II
methanotroph, a member of the *Alphaproteobacteria* that uses the serine pathway for carbon
assimilation *M. sedimenti* is a gammaproteobacterial type I methanotroph, using the ribulose
monophosphate (RuMP) carbon assimilation pathway (Tavormina et al., 2015). The pathway data
presented in Fig. 4 suggests that all methane-bound hydrogens are water exchangeable during the
catabolic oxidation of methane to carbon dioxide. Thus, to achieve a H:C tracer ratio less than 4, a





substantial proportion of methane-derived formaldehyde would need to proceed down the
assimilatory pathway, a requirement that was likely met given the cultures' increase in cell density.
The oxic incubations of methane seep sediment produced a H:C tracer ratio of 1.66 +/- 0.02
SE. Given that the known modes of biological methane oxidation – type I and type II aerobic
methanotrophy and reverse methanogenesis anaerobic methanotrophy – bound this observed value,
it appears likely that the oxic sediment incubations supported a mixture of both aerobic and
anaerobic methane oxidation processes. Aerobic methane oxidation likely dominated, based on the
$\sim 7 \times 10^4$ Pa partial pressure of $O_2$ and the proximity of the H:C tracer ratio to that of the aerobic
methanotrophic cultures, but anoxic niches likely remained or developed in the incubation bottles.
3.4. Validating the Monodeuterated Methane Approach: Anaerobic Methanotrophy at Pressure
To demonstrate the utility of the $CH_3D$ rate measurement approach in addressing
experimentally relevant questions, we sought to evaluate the influence of *in situ* pressure on
methanotrophic rates of Hydrate Ridge seep sediment microbial communities. Material collected
for microbiological studies of AOM is frequently obtained from marine settings of various depths
that are subjected to distinct and substantial pressure regimes (Ruff et al., 2015). Pressure is not
always rigorously incorporated into microcosm experiments, though evidence suggests it can be an
important determinant of methanotrophic rates (Bowles et al., 2011; Nauhaus et al., 2005; Zhang et
al., 2010).
Parallel seep sediment incubations were subjected to 0.1 MPa (atmospheric pressure) and
9.0 MPa (equivalent to ~900 m depth). Measured rates, expressed in δD values derived from D/H
ratios, are shown in Fig. 5. A significant increase in methane consumption was observed in both
live conditions at heightened pressure, corresponding to sediment incubated with isotopically
labeled glycine (samples 1a and 1b) and ammonium chloride (samples 2a and 2b). Controls



lacking CH$_3$D (samples 3a and 3b) and biological activity (samples 4a and 4b) showed no increase
in D/H ratios (see Table S2 for sample set-up details). The simulation of *in situ* Hydrate Ridge
pressures led to a 79.5% (+/- 6.5 SE) increase in relative methane oxidizing rates. Incubation with
500 μM glycine rather than ammonia at high and low pressures resulted in small but consistent rate
increases of 12% +/- 4.1% SE, potentially reflecting the energetic and biosynthetic distinction
between exogenous amino acids and unprocessed fixed nitrogen.

Previous reports have found a wide range of different pressure-related effects. In a sulfate-

coupled AOM bioreactor, pressures were varied from 1 to 8 MPa and sulfide production
approximately tripled, demonstrating Michaelis-Menten style kinetics with an apparent K$_m$ of 37
mM (Zhang et al., 2010). Methane partial pressures of 1.1 MPa led to a 5x increase in sulfate
reduction rates relative to ambient atmospheric pressure with Hydrate Ridge sediments
demonstrating methane-dependent sulfate reduction (Nauhaus et al., 2002). With methane seep
sediment from the Japan Trench, however, methane-driven sulfate reduction rates did not correlate
with changing pressure (Vossmeyer et al., 2012). Nauhaus et al. (2005) suggested that the pressure-
induced rate increases are due more to heightened methane solubility and bioavailability rather
than physiological effects or biomolecular re-ordering. Bowles et al. (2011) presented a very
different perspective by showing a six- to ten-fold AOM rate increase at 10 MPa when methane
concentrations were held constant. Deconvolving these two influences and how they depend on
community composition or physicochemical parameters is feasible with pressure chamber
experiments utilizing monodeuterated methane. Understanding the relative contributions of
environmental and physiological effects to methane oxidation will help constrain methane fluxes
across a larger envelope of the planet's methanotrophically active zones.
3.5. Monodeuterated Methane in Experimental Investigations



Based on $^{14}CH_4$ ground-truthing experiments with aerobic methanotrophic cultures, oxic

seep sediment, and anoxic seep sediment, as well as the proof-of-concept pressurized experiments,
we believe that the monodeuterated methane approach to methane oxidation rate measurement is a
useful addition to the biogeochemist's tool set. Compared with radiolabel approaches ($^{14}CH_4$, $^3$H-
$CH_4$, $^{35}SO_4^{2-}$), the method requires less safety-oriented planning, and is logistically simpler, more
affordable, and less susceptible to isotope fractionation effects. Our results suggest that it appears
to be a more precise method based on standard error calculations, though direct comparisons are
complicated by the fact that different aliquots of the same initial material were used. Because the
monodeuterated methane method focuses on methane-bound hydrogen atoms, it offers different,
complementary information about methanotrophic systems than carbon-based techniques like
methane or bicarbonate quantification. While this distinction complicates the interpretation of
isolated D/H ratios, it can offer an additional dimension of information for analysis of methane-
derived intermediates in relevant metabolisms. Given these caveats, we recommend three use cases
for monodeuterated methane in methane oxidation rate measurement applications.
1)    First, the approach can be employed in a strictly comparative context using analogous

inoculum exposed to a range of different conditions, as demonstrated with the pressure-

based sediment incubations presented above. Evaluating the effect of different

conditions such as temperature ranges, chemical concentrations, or energetic landscapes

on seep sediment methane oxidizing rates would all be promising applications.

Comparative analysis of AOM rates at different seep sites would also be useful,

provided anaerobic or aerobic methanotrophic processes could be isolated.

2)    Second, by performing side-by-side monodeuterated methane and radiocarbon tests, a

sample-specific H:C tracer ratio can be determined, and absolute rates of full methane



oxidation can then be inferred in subsequent experiments based exclusively on D/H
ratios. Conducting such paired studies under additional environmental or lab-based
conditions would help clarify the universality of the ratios presented here and would
likely reveal additional questions of metabolic dynamics in a range of experimental
systems.
3)  Finally, the use of monodeuterated methane as an analytical tool, alongside additional
methods such as carbon- or sulfur-tracking procedures, would enable a multi-
dimensional examination of anabolic and catabolic processes in methane-based
metabolisms. In particular, the H:C tracer ratios presented here reveal intriguing and
seemingly systematic relationships between carbon and hydrogen anabolic and catabolic
partitioning across distinct physiologies, yet an underlying theoretical framework
regarding the fate of methane-bound hydrogen atoms remains outstanding. In anaerobic
methanotrophic systems, back-reaction rates and equilibrium constants could be
evaluated by a) including a $^{13}CO_2$ signal in the water and measuring $^{13}CH_4$, and/or b)
utilizing multiply deuterated methane as initial headspace and measuring all possible
isotopologues via nuclear magnetic resonance (NMR) or high resolution mass
spectrometry. For aerobic methanotrophs, evaluating H:C tracer ratios under more
clearly defined growth and maintenance phases would elucidate distinct H:C tracer
ratios associated with catabolic, RuMP, and serine pathways, enabling future use of that
parameter as an arbiter of relative anabolic and catabolic activity. Furthermore,
additional environmental variables can be tested to gain insight into distinct redox
pathways and dynamics of reversibility. For example, AOM under lower sulfate
concentrations might be expected to generate higher H:C tracer ratios (Yoshinaga et al.,



2014), and this parameter could be further developed as a measure of microbially
mediated isotopic equilibration.

## 4. Conclusion

The ability to accurately measure methane oxidation rates – both comparatively and in
absolute values – is an important component of methanotrophic studies. Such measurements
frequently depend on radiotracers or measurements of chemical species that are related to, but not
directly indicative of, methane metabolism. The monodeuterated methane technique presented here
represents a novel approach to methane oxidation rate measurements, notable for its logistical and
analytical ease (particularly in ship-board applications), as well as the added dimension provided
by H-based, rather than C-based, information. We have demonstrated that the D/H ratio is directly
proportional to methane oxidation rates as measured in absolute terms by the well-established
$^{14}CH_4$ method. The value of the proportionality constant differs based on the experimental system,
likely dictated by relative proportions of aerobic and anaerobic methanotrophic metabolisms,
though additional experiments to determine the nature of the putative mixing line are needed.
Methane biogeochemistry is a dynamic field of study with implications for carbon cycling,
microbial ecology, and climate dynamics, though experimental challenges have slowed our
understanding of methane-based biological reactions. With the $CH_3D$ approach as an added tool in
the arsenal of rate-based examinations, a broader understanding of the intricacies of methane
metabolism, as well as its role in environmental and anthropogenic systems, is within reach.

## 5. Acknowledgements



We thank the Captains, Crew, *Alvin* group, *Jason* group, and Science party members from *RV Atlantis* legs AT-15-68, and AT-18-10. Water analyzer measurements were conducted in the laboratory of Alex Sessions at the California Institute of Technology with technical support from Lichun Zhang. We are indebted to William Berelson at the University of Southern California and Nick Rollins for use of their pressure chambers and assistance with the incubation experiments. We thank Alex Sessions, Woodward Fischer, Dianne Newman, Tori Hoehler, Amy Rosenzweig, and Daniel Stolper for helpful conversations during the preparation of the manuscript. This study was funded by grants from the U.S. Department of Energy, Office of Science, Office of Biological and Environmental Research (DE-SC001057), the NASA Astrobiology Institute (Award # NNA13AA92A) and support from the Gordon and Betty Moore Foundation through grant GBMF3780 (to VJO). JJM was supported by a National Energy Technology Laboratory Methane Hydrate Research Fellowship funded by the National Research Council of the National Academies. This research used resources of the Oak Ridge Leadership Computing Facility. Oak Ridge National Laboratory is supported by the Office of Science of the U.S. Department of Energy.

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

Table 1: A summary of the samples used for all experiments conducted in this study. Green boxes

indicate that the experiment took place (with all relevant permutations and controls, as described in

the text); blank boxes indicate experiments that were not conducted. $CH_3D$ refers to

methanotrophic rate experiments using the novel monodeuterated methane technique, while $^{14}CH_4$

refers to the radiolabel-based experiments. The three-part codes for samples derived from

environmental material refer to active (A) or low-activity (L) sediments (Sed) or carbonates (Carb),

as explained in the text.





| | | Oxic | | Anoxic | |
|---|---|---|---|---|---|
| | | $CH_3D$ | $^{14}CH_4$ | $CH_3D$ | $^{14}CH_4$ |
| Aerobic Methanotroph Cultures Experiment | | | | | |
| | *M. trichosporium* | | | | |
| | *M. sedimenti* | | | | |
| Seep Sediment Experiment | | | | | |
| | A.Sed-5128 | | | | |
| | L.Sed-5043 | | | | |
| Seep Carbonate Experiment | | | | | |
| | A.Carb-5305 | | | | |
| | A.Carb-5152 | | | | |
| | L.Carb-5028 | | | | |
| Pressure Experiment | | | | | |
| | A.Sed-3450 | | | | |


Table 2: H:C tracer ratios for the experimental treatments addressed in this study.

| **Aerobic Methanotroph Cultures** | | |
|---|---|---|
| | Exponential Phase | Stationary Phase |
| *M. trichosporium* | 1.5 | 1.48 |
| *M. sedimenti* | 1.54 | 1.59 |

| **Methane Seep Sediments and Carbonates** | | |
|---|---|---|
| | Oxic Incubations | Anoxic Incubations |
| A.Sed-5128 | 1.62 | 2.05 |
| L.Sed-5043 | 1.71 | 2.01 |
| A.Carb-5305 | 1.65 | 1.96 |
| A.Carb-5152 | 1.63 | 2.08 |
| L.Carb-5028 | 1.69 | 1.86 |



Fig. 1: Amount of methane oxidized over time for cultures of a) the type II methanotroph *M.*
*trichosporium* and b) the type I methanotroph *M. sedimenti* using the $CH_3D$ method (circles) and
the $^{14}CH_4$ method (diamonds), calculated as discussed in the text. Symbols correspond to sample
types as follows: blue = $CH_3D$-derived experimental data; brown = $CH_3D$ killed control data;
orange = $CH_3D$ abiotic control data; gray = $CH_3D$ oxygen-free, argon-infused control data; green =
$CH_4$ control data; red = $^{14}CH_4$ –derived experimental data; black = $^{14}CH_4$ –derived killed control



data. Error bars show standard errors for three biological replicates, with the exception of the $^{14}CH_4$
–derived killed control (n=1). Data obscured by other data series exhibited values between -60 and
110 nmol for a) and 0 and 60 nmol for b).

Fig. 2: Methane oxidation rates of a) oxic and b) anoxic incubations of active and inactive seep
sediment and carbonate rocks (n=3 in all cases). Values compare rates derived from the $^{14}CH_4$
(blue) and $CH_3D$ (green) experiments for a given sample material; standard error bars provided.

Fig. 3: A schematic diagram demonstrating the potential fate of methane-associated hydrogen
atoms in the "reverse methanogenesis" pathway. Hydrogen atoms are distinguished by color and
superscript number, and potential exchanges with inter- and intra-cellular water are shown.
Potentially detectable methane-derived hydrogen atoms (4, occurring throughout the oxidation
pathway) and carbon atoms (1, requiring full oxidation) are highlighted in orange and purple
boxes, respectively. Arrows are unidirectional to demonstrate the net methane-consuming direction
of the pathway, but all enzymes have been shown to be reversible (Thauer, 2008), a situation that is
shown explicitly only for Mcr. The extended dashed line represents the cell membrane.

Fig. 4: A schematic diagram demonstrating the potential fate of methane-associated hydrogen
atoms in the aerobic methanotrophy pathways. Hydrogen atoms are distinguished by color and
superscript number; asterisks represent location-specific ambiguity. Potentially detectable
methane-derived hydrogen atoms and carbon atoms are highlighted in orange and purple boxes,
respectively. Mmo enzymes are not believed to perform reversible reactions.



Fig. 5: Water δD values with standard error bars of seep sediment samples following 38-day
incubations with CH$_3$D at 9.0 MPa (brown bars, "b" samples) or 0.1 MPa (pink bars, "a" samples).



Figure 1

a)

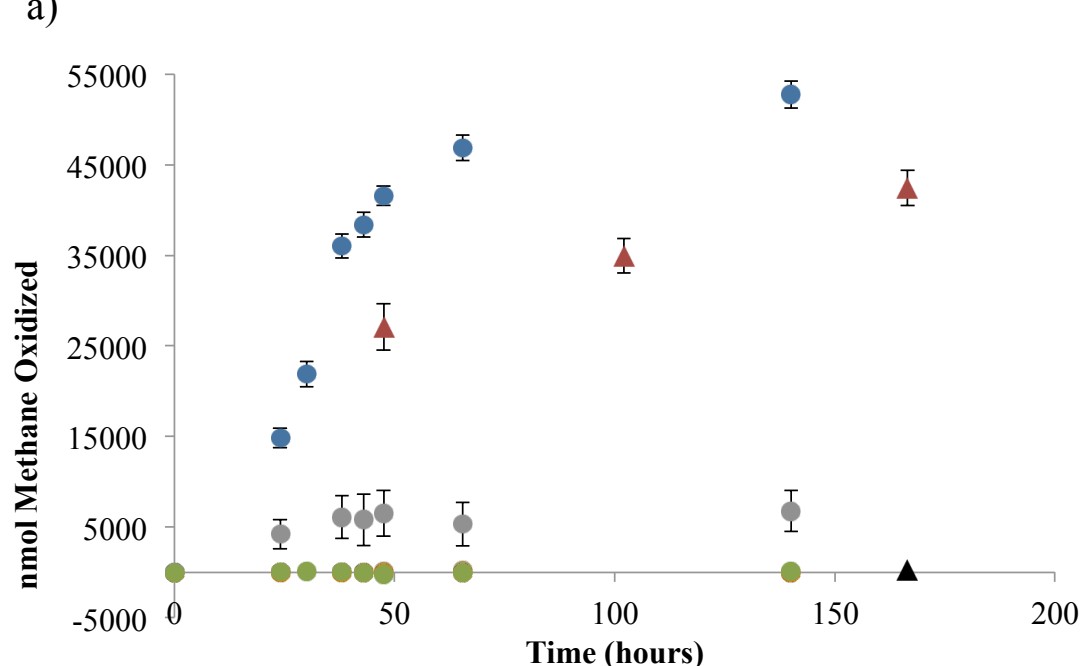

b)

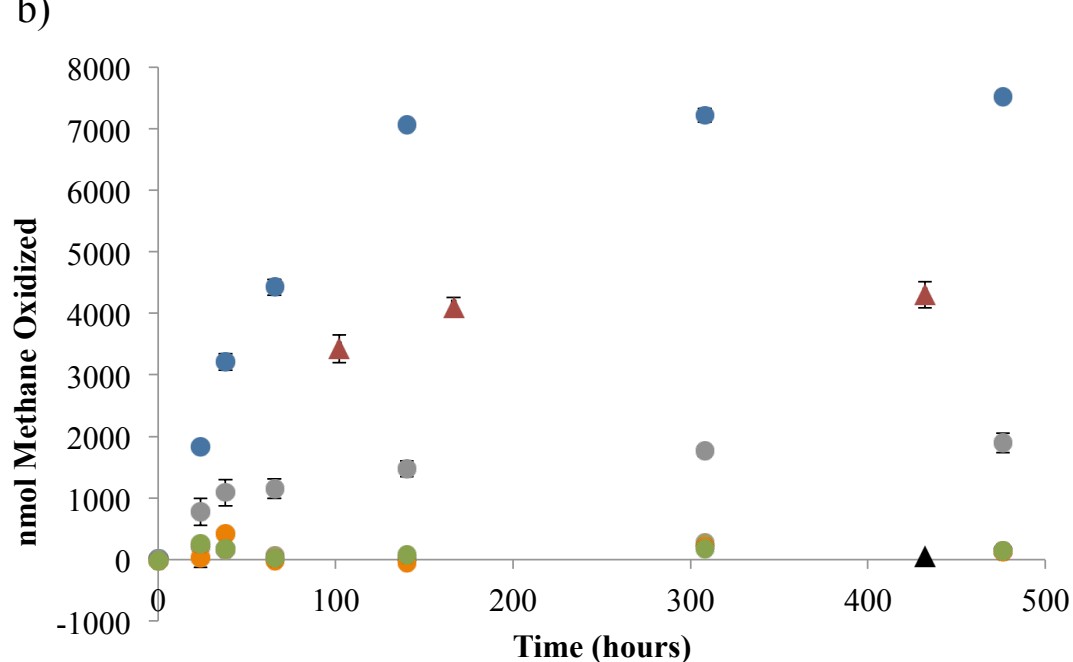





Figure 2

a)

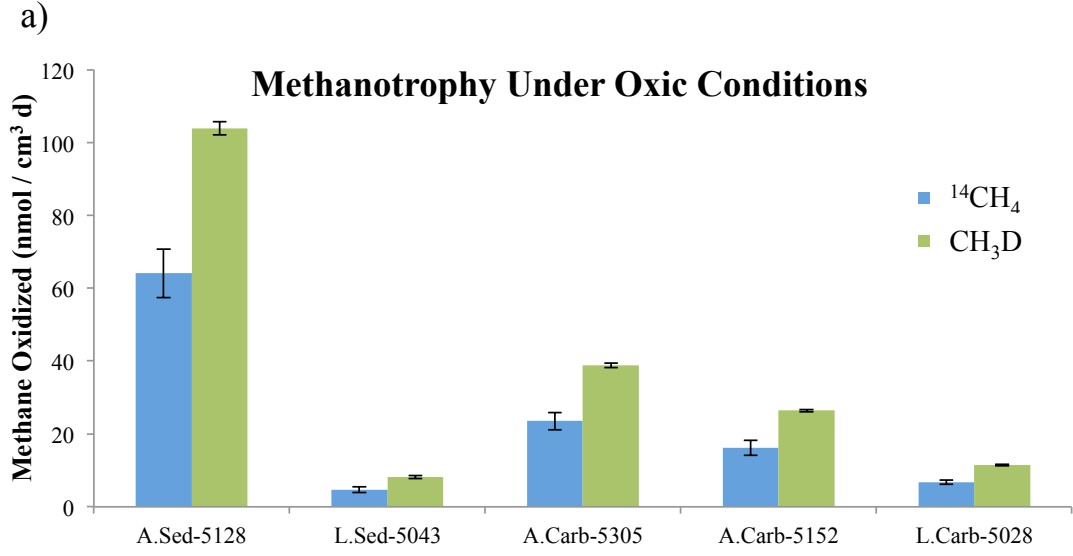

b)

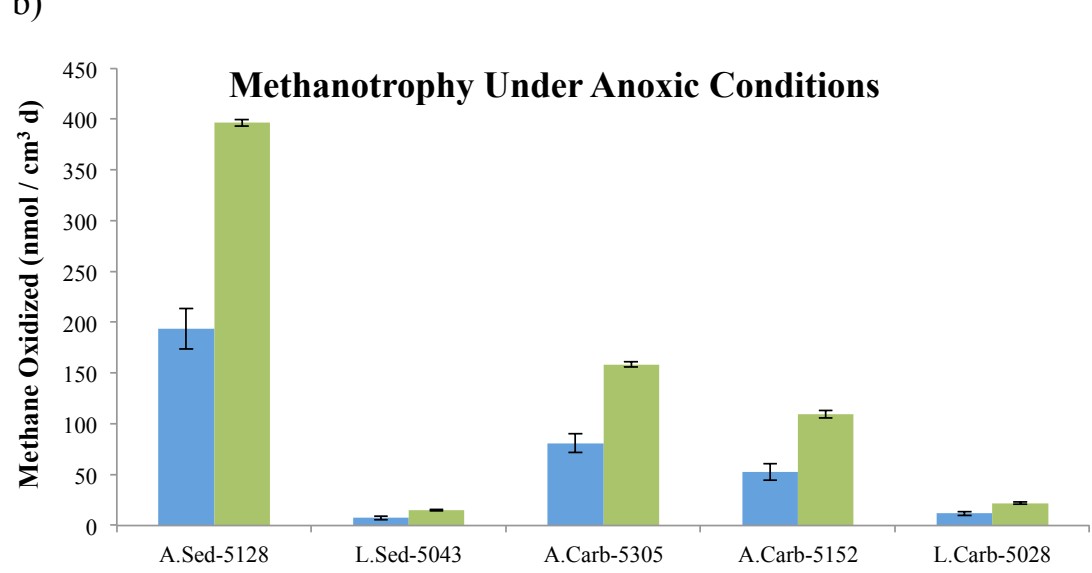



Figure 3





Figure 4





Figure 5

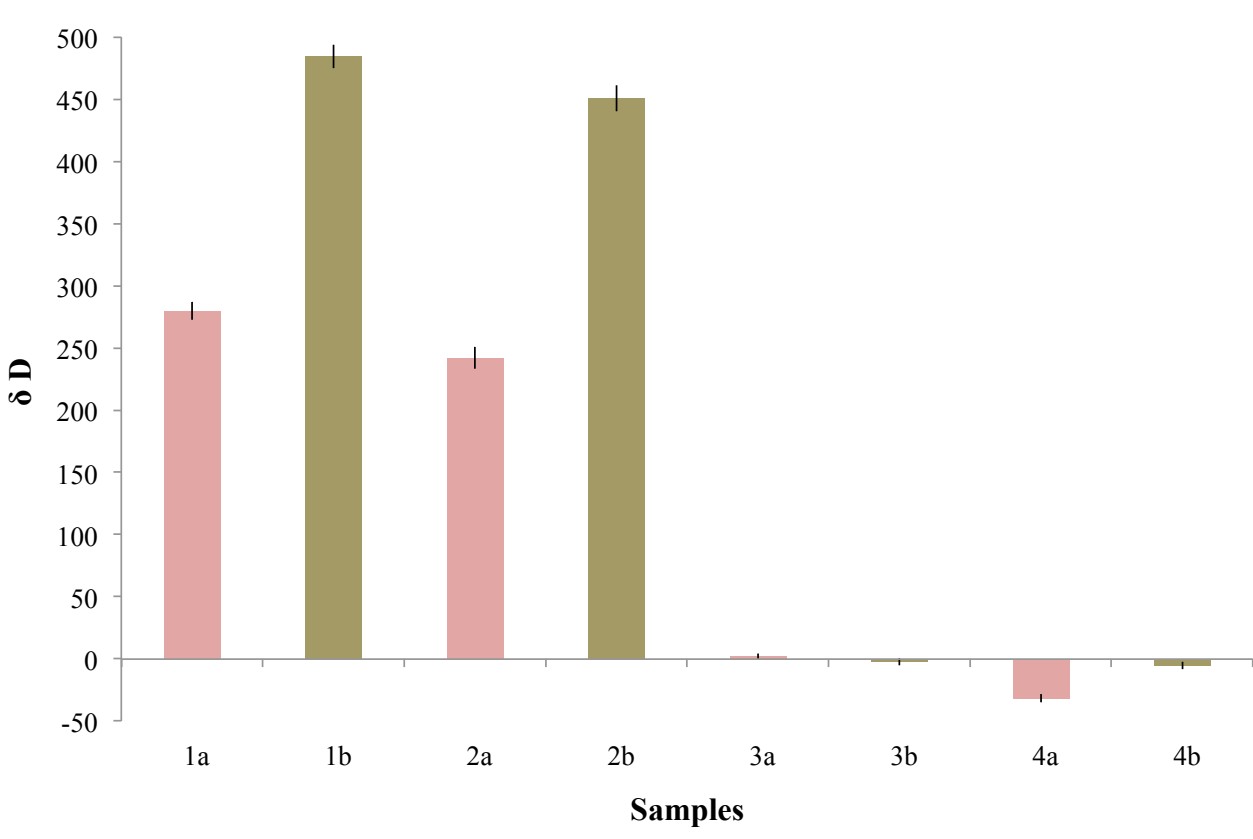