# Peer review of "MONODEUTERATED METHANE: AN ISOTOPIC PROBE TO MEASURE BIOLOGICAL METHANE METABOLISM RATES AND TRACK CATABOLIC EXCHANGE REACTIONS"

_Biogeosciences, 2016_

## Referee Comment (RC1) · Anonymous Referee #1 · 29 May 2016

Marlow and coworkers tested D-labeling of methane as an alternative tracer based method to determine methane oxidation rates in aerobic methanotrophic cultures, and in oxic and anoxic (AOM-active) sediments. D-labeled methane was added to a sample and the change of water deuterium isotopic composition was measured using Cavity-Ringdown Laser Absorption Spectroscopy (Los Gatos). Values were used to determine rates of methane oxidation. Alternatively the 14C-methane method was used and rates were compared. The authors found generally higher rates with the D-methane compared to 14C-methane tracer approach. As reasons for this, although not really clearly pointed, two mechanisms were discussed. Most aerobic methanotrophs use primarily methane as carbon source - hence 14C-methane is partly transferred into organic

biomass and not into CO2. Methane-H is instead almost fully transferred into water, hence seems to be promising for an accurate measurement of aerobic methanotrophy. In anaerobic methanotrophy back reaction that cause isotopic exchange were identified, which would lead to rather overestimated real rates. In general reported values have a good precision, meaning a low standard deviation – however it has not been tested if the rates are comparable to chemical measurements and as such lacks a trueness/ accuracy validation. The manuscript is full of slang, and often misses accurate terms, and lacks structure. Due to this in particular the discussion is not easy to follow. Furthermore section titles are not informative. This would need to be improved. Furthermore one has to criticize that the main, new method is insufficiently described. Why is there no formula? Instead the new approach is only described in text. From this I at least think that the rate determination is slightly incorrect. As far as I can anticipate the authors use ratios instead of (required) fractions in their calculation. Please also refer to the labeling percentage you used.

Below are my further comments, which are not complete – because sometimes I could simply not follow the discussions!

The abstract is not really productive, here a suggestion on how to improve it:

"Biological methane oxidation is a globally relevant sink of methane and it proceeds in aerobic and anaerobic pathways. However measuring rates of methane oxidation in natural samples remains challenging. Here we present a new approach for measuring rates of microbial methane oxidation that bases on the addition of labeled monodeuterated methane (CH3D) to a sample, and quantification of the label in water phase via isotope ratio mass spectrometry. As comparison we performed the well-established 14CH4 radiotracer approach the CH3D procedure. We provide measurement with cultures of I and type II aerobic methanotrophs and for sediment and carbonate rock samples incubated under anoxic and oxic incubation conditions. [results] [what is the different and why is it] We also employ this method to investigate the role of pressure on methane oxidation rates in anoxic seep sediment, revealing an 80% increase at the

equivalent of ~900 m water depth (40 MPa). [Conclusion ; where and why should the new method be used]

L47: Often involving ? where is the real counter evidence/ or do you mean the Haroon Paper?

L50: I would cite the Mc Glynn and Wegener nature studies on the mechanism for AOM, they seem to be the most promising approaches, also considering the new Scheller study Same Paragraph: As later aerobic methanotrophy is discussed it should also be introduced here

L56: The relevance in nature was discussed above, does not need to be repeated here

L58: AOM rate measurements have traditionally...

The performed measurement does not discriminate aerobic or anaerobic methane oxidation – I would rephrase it: The turnover of methane to carbon dioxide in environmental samples is often traced using stable or radiocarbon labeling approaches

L61: Stable isotope 13CH4 tracers is not correct: Methane labelled with 13C or something

L62: but the natural presence of 13C in marine dissolved carbon requires accurate detection of the reactant and product concentration and isotopic compositions. (I think that is what you mean.

L66: "though low molarities make samples susceptible to exsolution..." what should this mean? " I think you wanted to elaborate on concentration/ diffusion profiles.... But then you need to tell a full story... that it really not enough

L68: "carbon movement into oxidized species" .... Does not sound scientific – track the oxidation of methane to CO2

L69: Labeling with tritiated methane was introduced to track aerobic methane turnover in the water column. The shorter half-life time allows higher specific activity, and the

procedures for the separation of reactant and product are less complicated and can thus be performed on sea.

L75: measurements of methane turnover remain

L77: delete: "Nearly all of the aforementioned approaches are carbon-based; a hydrogen-based tracer offers an additional dimension to investigations of methane biochemical dynamics. "

L83: that aqueous D/H values are consistently proportional to 14C-based rate measurements for given laboratory treatments tested in this study

Of course it is the rates derived from the comparison of …. D/H are consistently proportional to those measured...

But better remove 82-92 : those are your results

Methods

2.1.1. Aerobic Methanotroph Cultures it is : Experiments with cultured aerobic methanotrophic bacteria After several successful transfers – how is a transfer successful – how did you see that cells were growing

L116: Parallel incubations incorporated 14CH4 to allow for three time points of destructive sampling, and killed and radiolabel-free controls. In parallel incubation we tested the turnover of 14CH4…. And then go on with a clear description as above

L121: Here you could have compared it with purely chemical measurements

2.1.2. Environmental Samples: Methane Seep Sediments and Carbonates Again that is not an informative headline: Measurement of methane turnover in seep sediments and seep carbonates

L137-46: Unnecessary

L192: Analytical procedures 2.2.1 Rename – that is not a CH3D rate measurement

Determination of rates based on deuterium oxide formation? Or something similar Bring also the D-Methane concentration first – or was this not done at all?

Methods: I don't see a formula how D-values are transformed into rates – even though it a new method which is describe. As a quick remark, to be mathematically correct and fully comparable, mathematically you cannot use ratios – you need to use fractions instead FD=R(DH)/(R(DH)+1) And please revise all your downstream calculations

How much of the methane was used up in total in your approach how much methane was added. Did you do any control measurement????

L234: Name in "Labeled 14C-inorganic carbon or define it once- it was before at pH14, so only CO3–.

L255: Your main new method is misplaced and only briefly described – you need to mention how you go from the spectra (which is the input of labeled methane) to the D2O

3. Results and discussion 3.1. "Aerobic methanotrophic cultures" is not a title that can refer to a result – name it Comparison of rate measurements in aerobic methanotrophic cultures 3.2. Similar as above 3.2 needs a complete revision – I don't understand it Since 3.1 and 3.2 are only results one could make them results, 3.3. could be discussion

3.3. It is not the "H:C tracer ratio" that needs understanding here, but the rates derived from 13CH4 and CDH3 incubations

"The CH3D and 14CH4 approaches quantify distinct aspects of methanotrophy,"??? I thought you mention both quantify turnover of methane?

L313: Change to: Because methane is an inert molecule abiotic exchange between methane and water protons is not expected, and indeed the D/H ratios in controls remained stable. Use term significantly/ significant should be reserved for statistical analyses. Not here. Please show those results and any measurement somewhere –

the reference to Fig.1 is inappropriate- I don't see isotope values there

L323: If you refer to backfluxes such as in Holler et al., you should allow the backfluxes throughout the model (figure 4). And based on this and further studies (i.e. Yoshinaga et al. ,2014) relative backfluxes in AOM should increase with lower / no methane

3.3.1. The H:C Tracer Ratio in Anaerobic Methanotrophy (Sorry, again this title is less than sloppy). Relation methane turnover rates derived from D and C labeling ? or something similar

L327-360 I cannot follow this discussion, especially from L332, I have a vague idea what the author wants to tell. The reversibility of specific steps, in particular of the activation step will lead to higher apparent rates in D compared to C labeling.

379: Quantifying anaerobic methane oxidation at different methane pressures

3.5 I would put the monodeuterated methane experimental investigations further up to the results. The interpretation of these experiments is really not well judged.

L421: Why do you think one would have less isotope fractionation effects in D than in C?

L422: The precision to meet a chemically measured value was unfortunately not tested at all – so the results might be highly reproducible but we don't know yet what they show.

L423ff: Please state what both methods achieve: In aerobic methanotrophy D labeling might be more realistic to reproduce total complete oxidation rates of methane than the methane carbon labeling, because methane carbon is partly assimilated. In AOM C-labeling should be better to track methane oxidation, as a backflux would be small. D-labeling of methane likely overestimates methane oxidation rates, as rate – determining backflux can occur at all steps.

4. Conclusion. You might have been precise with your measurement but accuracy or

even trueness of your measurement was not tested

---

## Referee Comment (RC2) · Anonymous Referee #2 · 8 Jul 2016

General comment

The manuscript describes a novel technical approach to measure biological methane oxidation and track H-atoms through methanotrophic metabolisms. In part one of the manuscript, the authors focus on a comparison between the new CH3D method with the established 14CH4 method. For comparison of the results of the two methods a scaling was determined. To evaluate their method the authors performed measurements on methanotrophic culures and environmental sediment samples. Part two describes how the CH3D approach can be used to track H-atoms in anaerobic an aerobic methanotrophic pathways. Part three deals with a pressure experiment in which sediment samples were incubated at 9 MPa and 0.1 MPa to discuss the effect of pressure

on methane turnover. The title of the manuscript describes the work adequately and the manuscript is well structure and written. Based on the few comments below, I suggest minor revision before the manuscript will be published.

Specific comments

Abstract

The abstract is well written and nicely reflects the outcome of the work.

Line 10. The poor precision of the established methods to determine a methane oxidation rate is mentioned (e.g. 14CH4). That is definitely true and an increase of precision is desirable. However, if precision is one of the major points that should be improved, the authors should tell the reader something about how (and in which range) the new method affect the precision of the measurements. That should be integrated in the abstract with a comparison in numbers of the precisions "14CH4 against CH3D" - derived from their comparative studies.

Line 14. The description of the central analytical device is rather weak "...using a water isotope analyzer". Since this is the new basic tool that allows this new approach, a subclause about how the system works (Off-axis ICOS technology) should be already integrated in the abstract.

Line 21ff. The pressure experiment is very nice approach that shows how important incubations under in situ conditions are to determine real methane turnover rates. However, the story get lost in the abstract (just 2 lines) and appears a bit out of context (see comments below to chapter 3.4).

Line 26ff. Point 2 is difficult to understand without reading the manuscript. The phrase "scaling factor" is difficult to understand without a deeper context. I would suggest reformulating and extending the sentence.

Introduction

The introduction is well structured and gives appropriate background information. Line 46-55. AOM is introduced but no background information about aerobic methane oxidation (MOB types) are delivered. This should be completed, because the lab test presented in the manuscript are not only focused on AOM.

Line 57. "biogeochemical curiosity", please rephrase or explain what you mean in more detail.

Line 71. References. I would also add a more recent paper, because the methods used changed a bit (e.g. I. Bussmann et al., Assessment of the radio 3H-CH4 tracer technique to measure aerobic methane oxidation in the water column, L&O Methods, 2015

Line 84ff.Why did the authors decided to test their new approach against 14CH4 and not also against tritium labeled methane with its improved specific activity that allows incubations under more realistic methane concentrations.

Line 85. Please explain in more detail what you mean with "partial versus complete methane oxidation".

Line 91ff. This sentence is redundant and could be deleted.

Line 82ff. I cannot find any hint to the pressure experiment in this outlook. Such an outlook should cover the main aspects discussed in the following text.

Methods

The chapter is well structured and explains the different methods in an appropriate way. Line 100ff. As mentioned above, also here the pressure experiment is a bit out of context. Why did the authors decide to perform these experiments without a comparison with 14CH4 rate measurements? See also comments below for chapter 3.4.

Line 130. Is the information about "unique four-digit serial number" needed? I think the

sentence can be deleted.

Line 131. Maybe insert: "The "active" designation in our sample description (e.g. Figure 2) refers..."

Line 141. That is difficult for me to follow. A few sentences before the authors say that carbonates are formed during "active" periods of seepage (line 137) and now they say that carbonates (L. Carb) can also exist under "low-activity". Please explain the difference between L. Carb and A. Carb in more detail.

Line 147ff. I would suggest to move the entire paragraph to line 130. First describe how the samples were taken and then how the samples were named (paragraph 130ff).

Line 153. The samples were stored in Ar-flushed bags. Does this influence the methane concentration in the sample and also maybe the activity of the microorganisms? How long were the sample stored?

Line 155. The samples were maintained under 2x10ˆ5 Pa CH4 headspace for one month. Why one month? And how does that fit to in situ conditions (methane concentration)? And if there are differences can we expect that it also influences the activity of microorganisms in the experiment? A comment on that should at least be given in the discussion somewhere.

Line 166. Which gas was injected – CH4? And why does it end in a desirable headspace composition?

Line 168. What was the reason to choose this specific gas composition? Does it reflect environmental conditions?

Line 180. What are "mylar" bags. Are they gas tight? Maybe a short comment on that in the text.

Line 180. Actually, I could not find Table S2 in my documents and therefore cannot comment on that.

Line 188. Why did the authors choose 9.0 MPa. Where does the sample come from (water depth, temperature). Some more comments on the sample are needed.

Line 189. The authors tested visually the bags for leaks. I think a better method to test for leaks would be the analyses of CH4 (or CH3D) in the water of the pressure vessel at the start point and end point of the experiment.That would also deliver information about diffusion of methane through the bag into the surrounding water. Can diffusion be excluded?

Line 194. How was the volume of the water sample replaced in the culture?

Line 196. The only information about the main analytical device is the name of the model and the company. Since this tool represents something that is really new in the context of methane rate measurements, I would like to have some more details about the main analytical principle of the system (Off-axis ICOS technology). Can the authors deliver any additional references to the system (other studies)?

Line 205ff. What does it mean "sub-optimal". Is a statistical test behind that?

Line 211. I am not sure if I understood this part correctly. Is the assumption of a linear scaling factor only based on two standards? Is the LGR system linear over the measurement range? Was this tested?

Results and Discussion

Figure 1. It would be easier for the reader to follow the discussion, if Fig. 1a and 1b would be tilted with the name of the two MOB. I would also add a legend into the figures to explain the different symbols. The axis labels and the numbering on the axis do not look very accurate: the positions of the axis labels at the y-axis is not centered in both figures; 0 on the x-axis cuts the y-axis.

Line 271. "Using data points..." please list the data points used to derive the ratio in the text. Not only time also the methane oxidation rates.

Line 281. To which table or figure do these numbers (e.g. #1b) belong to?

Line 284. Does the 14CH4 method yield the "full-oxidation methanotrophy"? I think the correction of the CH3H oxidation rates using the H:C tracer ratio can just deliver oxidation rates, which can be better compared with the rates obtained with 14CH4 rate measurements.

Line 291. Please specify what you mean with "second time point"? IN which table or figure can I find the numbers 4d or 8d?

Line 371. Is any data available from the experiments to determine the cell density?

Chapter 3.4 As mentioned before, I have the feeling that this part of the manuscript is a bit out of the main focus. It is for sure an interesting approach but if this approach would be extended (more samples, different simulations (e.g. pressure),...), it could stay for itself. My main question are: What is the goal of these pressure studies? To show that pressure influences methane turnover? What is the advantage of the CH3D method compared with the 14CH4 method for these kind of pressure experiments? I am sure that the influence of in situ pressure is more important for the outcome of the experiment than the use of the new CH3D rat measurement approach (e.g. higher precision?). I think it must be explained in more detailed why exactly such an experiment can help to evaluate the now CH3D approach (without having data from a parallel 14CH4 approach). Line 392. Isotopically labeled glycine and ammonium chloride was not mentioned before in the manuscript. Please give detailed information about this experiment already in the first part of the manuscript (e.g. paragraph 82ff). For what is good for? What is the goal of that labeling experiment?

Line 417. Please explain why the pressure experiment is a proof-of-concept. Pressure makes the difference in this experiment not the method that was used for methane oxidation rate measurements.

Line 437. One advantage of the CH3D method is that "it does not require the logistical, safety, and administrative hurdles associated with radiotracers such as 14CH4..."
(line88ff). But to obtain absolute rates of full methane oxidation, parallel incubations
with CH3D and 14CH4 must be performed. That means that we still have to take ra-
diotracer on ships (together with the CH3D lable and analytical equipment) with all the
administrative hurdles. That means no advantage for expeditions?

Figure 1 See comment above Colors are difficult (e.g. I cannot see brown on my
printout). Would suggest to change the colors. Capture:

Figure 3 and 4 Please give the figure titles like "anaerobic methanotrophy pathway"
and "aerobic methanotrophy pathway".

Figure 5 A legend (and also a title) in the figure would be helpful. Capture: That the
data comes from the pressure experiment should be mentioned.

---

## Referee Comment (RC3) · Anonymous Referee #3 · 22 Jul 2016

The authors tested a potential new method to use monodeuterated methane ($CH_3D$) as a metabolic substrate for methane oxidation rate measurements by quantifying the change in the aqueous D/H ratio over time. In the study, two methanotrophic cultures and several environmental samples were used to compare the radio of the novel $CH_3D$ method with the existing and well established 14C isotope method. The new approach is complementary to existing radio (14C)- and stable (13C) carbon isotopic methods. Because it isn't a stand-alone technique; stressing the alleged advantage of a non-toxic, rapid, and easy-to-use method is obsolete.

The overall structure of the manuscript is clear, focusing on the different metabolic pathways, which were approved by the new method by using different biological samples.

The authors give a nice overview of the potential of the presented method, although the argumentation in parts of the manuscript isn't easy to follow. From my perspective, the principle of the method isn't sufficiently explained (see comments below as well as the statements of the other referees). Also the not informative subtitles need to be improved!

I suggest minor revision before the manuscript will be published.

The abstract is smooth to read and well structured.

The introduction gives a nice overview of the main topics of the publication and is well structured. L36 the biogeochemical cycles in BOTH natural freshwater and marine environments is mentioned, but in L39 only the estimated methane emission in marine settings is given. Can you show values of produced methane in natural freshwater and rice fields/wetlands/permafrost as well?

L47 biochemical intricacies – what is meant with that phrase?

L48 -55 AOM is described. Because aerobic methane oxidation is a major part of the experimental setup and the discussion, aerobic methane oxidation should be introduced as well.

L57 biochemical curiosity – what precisely is meant by this?

L58 AOM rate measurements have traditionally been conducted using a handful of techniques. – I understand this sentence as introduction for the following methods, which not discriminate aerobic or anaerobic methane oxidation. It should be rephrased, see RC1.

L61 Stable isotope 13CH4 tracers – 13CH4 isn't a tracer. Better: 13C-labelled methane

L70 and the procedural advantages of working with a water-phase product rather than gaseous products – this is one of the main advantages? I think RC1 makes a good point.

L107 and 109 where did you get the cultures from? Are these maintenance cultures or ordered as pure culture from a company?

L118 how did you see that the exponential growth phase was reached? Which test did you use?

L130 All samples received a unique four-digit serial number. – Unnecessary.

L151 What is "compacted sediment"? And would it be interesting to know from what depth below seafloor (which layer of the push core) the sediment comes from? Or is it the whole push core sediment, which was transferred into a bag? But still, from what depth below seafloor comes the sediment?

L152 What's a mylar bag? There is a reference for the glass bottle (L158) but not for the bag. In general: check references for lab equipment.

L171 1.9 days? Better use hours, if it is necessary to mention the exact time point.

L190 how did you do the leak check?

L198 the water isotope analyzer determine the D/H ration of the sample – can you explain more detailed how the analyzer works?

I am agree with both referees: a formula would be helpful to understand the method and the principle of the general measurement procedure. Maybe it is possible to visualize both in a schematic diagram.

L281 tubes #1a, #1b, and #1c – from which tubes/samples are you talking about? I assume you mean your replicates, then either delete the specification of tube labels and just talk about replicates or expand table 1 and include that kind of extra information.

Chapter 3.4. I am not sure, if the pressure experiment gives an additional value to the manuscript or rather create more confusion. The main goal is to present the method and to explain the method in a way that it's clear and easy to understand – that is sometimes a challenge especially in Chapter 3.3.

L454 is, where you bring up by the first time in the whole article for what the abbreviation NMR stands for. Should be mentioned earlier in L256.

---

## Author Comment (AC1) · 18 Aug 2016

MONODEUTERATED METHANE:
AN ISOTOPIC PROBE TO MEASURE BIOLOGICAL METHANE METABOLISM
RATES AND TRACK CATABOLIC EXCHANGE REACTIONS

Jeffrey J. Marlow[1*#], Joshua A. Steele[1^], Wiebke Ziebis[2], Silvan Scheller[1],
David Case[1], Victoria J. Orphan[1]

[1] Division of Geological and Planetary Sciences, California Institute of Technology, Pasadena, CA, 91125 USA
[2] Department of Biological Science, University of Southern California, Los Angeles, CA, 90089 USA
* Current address: Dept. of Organismic and Evolutionary Biology, Harvard University, Cambridge, MA 02138 USA
^ Current address: Southern California Coastal Water Research Project, Costa Mesa, CA 92626 USA
**Correspondence email: marlow@fas.harvard.edu**

**Abstract**

Biological methane oxidation is a globally relevant process that mediates the flux of an important greenhouse gas through both aerobic and anaerobic metabolic pathways. However, measuring the rates of these metabolisms presents many obstacles, from logistical barriers to regulatory hurdles and poor precision. Here we present a new approach for measuring rates of microbial methane metabolism, using monodeuterated methane ($CH_3D$) as a metabolic substrate and quantifying the change in the aqueous D/H ratio over time using off-axis integrated cavity output spectroscopy. This method represents a non-toxic, comparatively rapid and straightforward approach that is complementary to existing radio- ($^{14}C$) and stable ($^{13}C$) carbon isotopic methods; by probing hydrogen atom dynamics, it offers an additional dimension through which to examine the rates and pathways of methane metabolism. We provide direct comparisons between the $CH_3D$

procedure and the well-established $^{14}CH_4$ radiotracer method for several methanotrophic systems, including type I and type II aerobic methanotroph cultures – for which the new approach is five times more precise – and methane seep sediment and carbonate rocks under anoxic and oxic incubation conditions. We also employ this method in a non-traditional experimental set-up, investigating the role of pressure on methane oxidation rates in anoxic seep sediment. Results revealed an 80% increase in methanotrophic rates at the equivalent of ~900 m water depth (40

MPa), revealing an important environmental parameter for methane metabolism and exhibiting the flexibility of the newly described method.

The monodeuterated methane approach offers a procedurally straightforward, reliable method that advances three specific aims. First, it allows users to directly compare methanotrophic rates between different experimental treatments of the same inoculum. Second, by empirically linking the $CH_3D$ procedure with the well-established $^{14}C$-radiocarbon approach, an absolute scaling factor can be determined for new systems of interest. This "ground-truthing" strategy enables "CH$_3$D only" experiments to yield rates of full methane oxidation; we demonstrate this principle in the context of several methanotrophic systems. Finally, monodeuterated methane facilitates a continued evaluation of C- and H-atom tracking through methanotrophic metabolisms, with specific foci on enzyme reversibility and anabolic/catabolic branch points. The procedural advantages, consistency, and novel research questions enabled by the monodeuterated methane method should prove useful in a wide range of culture-based and environmental microbial systems to further elucidate methane metabolism dynamics.

**1. Introduction**

Methane-consuming microbial processes represent an important component of biogeochemical cycles in natural freshwater and marine environments, as well as in human- impacted systems. In terrestrial soils, methane production in rice fields, anoxic wetlands, and thawing permafrost supports methanotrophic communities (Holzapfel-Pschorn et al., 1985;

Mackelprang et al., 2011). In marine settings, an estimated 85 Tg of methane per year, derived from biogenic and thermogenic sources, enters the subseafloor, the vast majority of which is anaerobically consumed in anoxic sediments (Reeburgh, 2007). Much of what remains is taken up in microoxic or oxic zones of the sediment or water column by aerobic methanotrophic microorganisms (Valentine et al., 2001). In freshwater wetlands, approximately 200 Tg of methane is generated per year, most of which is oxidized by hydroxyl radicals in the troposphere (Kirschke et al., 2013). Methanotrophy is also of interest in a range of human-impacted contexts, including wastewater treatment plants (Ho et al., 2013), landfills (Scheutz et al., 2009), and oil spills (Crespo-Medina et al., 2014).

In addition to the climatic and economic implications of the methanotrophic process, its biochemical details have stimulated many investigations. The anaerobic oxidation of methane (AOM) has proven particularly enigmatic, often involving a mutualistic relationship between anaerobic methanotrophic (ANME) archaea and sulfate reducing bacteria (SRB; Boetius et al.,

2000; McGlynn et al., 2015; Scheller et al., 2016; Wegener et al., 2015). A consensus on the precise nature of the mutualism remains outstanding, but the net result of the process is typically the stoichiometric oxidation of methane coupled with sulfate reduction (Knittel and Boetius, 2009).

Alternative electron acceptors including nitrate (Haroon et al., 2013), and nitrite (Ettwig et al.,

2010) have been demonstrated, while several studies have presented equivocal evidence for methane oxidation coupled directly to iron or manganese reduction (Beal et al., 2009; Nauhaus et al., 2005; Sivan et al., 2014).

Methane is oxidized aerobically by members of the classes *Gammaproteobacteria* (Type I)

and *Alphaproteobacteria* (Type II); verrucomicrobial representatives were more recently found to perform aerobic methanotrophy under extremely acidic conditions (Dunfield et al., 2007; Op den

Camp et al., 2009). Methane monooxygenase converts methane to methanol, which is further oxidized to formaldehyde; assimilatory pathways branching at this point can incorporate carbon into central metabolism through the ribulose monophosphate (RuMP) cycle (Type I

methanotrophs) or the serine cycle (Type II). Remaining formaldehyde can undergo two additional oxidation reactions, being converted first to formate and ultimately to carbon dioxide (Hakemian and Rosenzweig, 2007).

Methanotrophy is both a biogeochemically relevant force that modulates global climate and a poorly understood biochemical process; given this dual role, there is substantial interest in measuring the rate of the process and understanding elemental flows through metabolic pathways.

The oxidation of methane in environmental samples has traditionally been studied using a handful of techniques. Numerical models incorporating environmental sediment profiles of sulfate and methane concentrations can be used to back-calculate methane consumption rates (Jørgensen et al.,

2001). Methane labeled with $^{13}$C can be used to probe longer-term rates in controlled conditions (Moran et al., 2008), but the presence of natural $^{13}$C in marine dissolved inorganic carbon pools requires long incubations as well as accurate measurements of concentrations and isotopic compositions of reactants and products (Pack et al., 2011). Gas chromatography quantification of dissolved (Girguis et al., 2003) or headspace (Carini et al., 2003) methane concentrations has also been demonstrated as a rate measurement tool, though low concentrations can hamper reproducibility and exacerbate background contamination issues, particularly in field-based settings (Magen et al., 2014). Perhaps the most sensitive approach uses radiolabeled $^{14}CH_4$ to track the oxidation of methane-associated carbon to inorganic carbon species (Alperin and Reeburgh,

1985; Treude et al., 2003). Labeling with tritiated methane was introduced for water column aerobic methane oxidation measurements due to its higher specific activity and the procedural advantages of working with a water-phase product rather than gaseous products (Bussmann et al.,

2015; Valentine et al., 2001). Logistical challenges and health and safety regulations led Pack et al.

(2011) to develop an accelerator mass spectrometry detection method that requires $10^3$-$10^5$ less radiolabel than previous $^{14}$C and $^3$H approaches, though the analytical procedure remains labor intensive.

Despite the range of methods available, measurement of microbial methane utilization rates remains cumbersome, and the demonstration of a precise, safe, and easily enacted approach would be a welcome contribution for a diverse field of researchers. Nearly all of the aforementioned approaches are carbon-based; a hydrogen-based tracer offers an additional dimension to investigations of methane biochemical dynamics. Here we introduce a novel method for biologically mediated methanotrophy rate measurement that utilizes monodeuterated methane ($CH_3D$) as a substrate and measures the D/H ratio of the aqueous solution.

We demonstrate, through methanotrophic cell cultures and microcosm incubations of seafloor sediment and carbonate rock fragments, that methane activation rates derived from aqueous D/H values are consistently proportional to [14]C-based methane oxidation rate measurements for the laboratory treatments tested in this study. The resulting ratios, when viewed in the context of partial (methane-associated hydrogen exchange) versus complete methane oxidation (methane oxidation to $CO_2$), represent a new tool with which to examine the reversibility and catabolic / anabolic partitioning of methanotrophic metabolisms. As a proof of concept, we apply the monodeuterated methane approach to pressurized methane seep sediment incubations in order to test the role of an understudied environmental variable in methanotrophic rates under non- traditional empirical conditions. As a rate measurement protocol, this approach offers several advantages over current techniques: it does not require the logistical, safety, and administrative hurdles associated with radiotracers such as [14]$CH_4$ and [3]H-$CH_4$, it is less susceptible to analyte loss than methane headspace measurements, and compares favorably in terms of equipment cost and portability.

**2. Methods**

2.1. Experimental Set-Up

To demonstrate the precision and reproducibility of the monodeuterated methane approach, it was tested alongside the well-established [14]$CH_4$ radiotracer protocol. The use of [14]$CH_4$ is an accepted standard procedure in studies of methane consumption quantification (e.g., Knittel and

Boetius, 2009; Ruff et al., 2016; Segarra et al., 2013) and has been experimentally cross-referenced with methane concentration measurements (Treude et al., 2003) and other approaches including tritiated methane techniques (Mau et al., 2013; Pack et al., 2011). Both techniques were applied to a) aerobic methanotrophic cultures of *Methylosinus trichosporium OB3b* (kindly supplied by

Marina Kalyuzhnaya and Mary Lidstrom) and *Methyloprofundus sedimenti* (isolated from a deep sea whale fall; Tavormina et al., 2015)  b) oxic incubations of methane seep sediment and carbonate rocks, and c) anoxic incubations of methane seep sediment and carbonate rocks. In addition, the monodeuterated methane protocol was employed in a pressure-based experiment to demonstrate the technique's adaptability to distinct empirical set-ups and examine the relative effect of heightened, environmentally relevant pressure on methane consumption rates in anoxic seep sediment samples. Monodeuterated methane for all samples was 98% pure $CH_3D$ obtained from Sigma-Aldrich ($247 / L). For a representation of all experiments conducted in this study, see

Table 1.

*2.1.1. Experiments with Aerobic Methanotroph Cultures*

Cultures of *Methylosinus trichosporium* strain OB3b (Whittenbury et al., 1970) were grown using Nitrate Mineral Salts (NMS) medium at 30 °C. The newly characterized

*Methyloprofundus sedimenti* strain WF1 was grown in a modified NMS medium at 25 °C

(Tavormina et al., 2015). In both cases, shaking cultures were grown up from stock in sealed 25

mL test tubes that contained 5 mL media and 50:50 air:methane by volume. After several successful transfers (as determined by an increase in optical density, data not shown), experiments were initiated by passaging 0.94 mL of exponential phase inoculum into 8.5 mL media, for each of ten different experimental conditions, each prepared in triplicate (see Table S1). Due to the destructive nature of the $^{14}CH_4$ method, three of these triplicate sets were used to measure methane oxidation at three distinct time points.

Samples for D/H analysis were taken at seven time points throughout 140-hour (*M.*

*trichosporium*) and 476-hour (*M. sedimenti*) experiments. Sampling points were most concentrated around anticipated exponential growth phases as determined by optical density (600 nm) profiles of earlier rounds of culture transfers. Samples for radiolabel processing were taken at 46, 102, and

166.5 hours for *M. trichosporium* cultures and 102, 166.5, and 432 hours for the slower-growing

*M. sedimenti* cultures. Killed, cell-free, oxygen-free, and $CH_3D$-free controls were all assessed (Table S1).

*2.1.2. Experiments with Environmental Samples: Methane Seep Sediments and Carbonates*

Samples recovered from the Hydrate Ridge methane seep system were used to comparatively examine the novel monodeuterated methane ($CH_3D$) approach alongside the $^{14}CH_4$

protocol with environmental samples. Hydrate Ridge, Oregon, is located along a convergent tectonic margin and is well established as a site of methane seepage and sediment-based AOM

(e.g., Suess et al., 1999; Treude et al., 2003; Tryon et al., 2002). Methane concentrations within the most active seep sediments reach several mM, and have been measured and modeled at values up to 70 mM (Boetius and Suess, 2004) and 50 mM (Tryon et al., 2002) respectively.

Samples were collected with the Deep Submergence Vehicle (DSV) *Alvin* during *Atlantis*

leg AT-16-68 in September 2010 and the Remotely Operated Vehicle (ROV) *Jason* II during

*Atlantis* leg AT-18-10 in September 2011; materials used for methanotrophic rate experiments are specified in Table 1. The "active" designation in our sample descriptions refers to sites where methane seepage was manifested by seafloor ecosystems known to be fueled by subsurface methane (e.g. clam beds and microbial mats) or methane bubble ebullition. The term "low activity"

references sampling sites that did not exhibit any clear signs of contemporary methane seepage or chemosynthetic communities, though a small amount of methane supply and methanotrophic potential cannot be ruled out as subsurface advective flow can shift with time (Gieskes et al., 2005;

Marlow et al., 2014; Tryon et al., 2002). Sample types are abbreviated by the A.Sed (active sediment), A.Carb (active carbonate), L.Sed (low-activity sediment), and L.Carb (low-activity carbonate) designations. Seven samples were analyzed to examine a range of physical substrate type (sediment vs. carbonate rock) and seepage environments (active and low-activity): A.Sed-

5128, A.Carb-5305, A.Carb-5152, L.Sed-5043, L.Carb-5028, and sterilized control aliquots of

A.Sed-5128 and A.Carb-5305. Carbonate samples include both porous materials with macroscale vugs and pore spaces, as well as massive lithologies with more homogenous structure.

Shipboard, push cores and bottom water-submerged carbonates were immediately transferred to a 4 °C walk-in cold room and processed within several hours. To prepare material for future experimentation, sediment and carbonate rocks were stored in anoxic, Ar-flushed, gas-tight mylar bags (Impak Corp., Los Angeles, USA) at 4 °C until use several months later. In advance of experimental set-up, carbonate samples and homogenized sediment from the 0-12 cm push core horizon were prepared under anoxic conditions using 0.22 μm-filtered, anoxic $N_2$-sparged Hydrate

Ridge bottom water (at a 1:2 sediment/carbonate:bottom water ratio by volume). Samples were maintained under a $2x10^5$ Pa $CH_4$ headspace for one month in order to resuscitate activity; the corresponding dissolved concentration (3.7 mM) is consistent with environmental methane concentrations at Hydrate Ridge (Boetius and Suess, 2004).

To set up the experimental incubations, 10 mL physical substrate (compressed sediment or carbonate rock) and 20 mL filtered Hydrate Ridge bottom water were placed in 60-mL glass bottles (SVG-50 gaschro vials, Nichiden Riku Glass Co, Kobe, Japan). In all experiments involving carbonates, interior portions (> 5 cm from the rock surface) were used in order to ensure that properties exhibited were representative of bulk rock material and not a reflection of surface- based adherent cells or entrained material. Carbonate rock samples were fragmented in order to fit through the 28-mm diameter bottle opening; pieces were kept as large as possible to minimize the increase in surface area-to-volume ratio and maintain conditions as representative of the *in situ*

environment as possible. All bottles were sealed with butyl stoppers; following several minutes of flushing with $N_2$ (g), the headspace was replaced with methane, and an additional 30 mL of gas, whose composition varied depending on the experiment, was injected into the 30 mL headspace to generate an absolute pressure of approximately $2x10^5$ Pa. Anoxic incubation headspace was 100%

methane; oxic incubation headspace was 30 mL methane, 20 mL $N_2$, and 10 mL $O_2$. All incubation set-up prior to gas flushing and headspace injection took place in an anaerobic chamber. Triplicate samples, including killed controls, were prepared for all sample types. Measurements were taken for both D/H and $^{14}$C analysis at 46 and 96 hours for oxic incubations, and 72 and 192 hours for anoxic incubations. Anoxic active methane seep sediment (A.Sed-5128) incubations were used for nuclear magnetic resonance (NMR) analysis of the remaining methane (set up in triplicate, with 60

mL $CH_3D$ initial headspace) as well as empirical resolution studies sampled between days 20-22.

*2.1.3. Experiments with Environmental Samples in Pressure Vessels*

In order to probe the effect of pressure on anaerobic methanotrophic rates, a set of experiments was established, using the monodeuterated methane technique to determine relative rate differences. Active sediment from Hydrate Ridge (A.Sed-3450) was collected from a water depth of 850 m and an ambient temperature of 4 °C, processed shipboard, and prepared for experimentation as described above. To set up the incubations, eight 100 mL mylar bags were prepared with the components shown in Table S2 using homogenized sediment from the 0-12 cm horizon. 500 µM glycine or 500 µM ammonium were added in order to evaluate relative rate differences associated with organic and inorganic sources of nitrogen. Identical sets of four compositionally distinct samples – including killed controls – were established such that each treatment could be subjected to low pressure (0.1 MPa, i.e., atmospheric pressure) and high pressure (9.0 MPa, equivalent to ~900 m water depth). Prior to gas addition, each bag was flushed for 5 min. with Ar.

The use of flexible mylar bags is essential for the application of external pressure, yet it presents obstacles for "traditional" methanotrophic rate measurement protocols such as the $^{14}CH_4$

method. In particular, the processing of post-incubation headspace is optimized for stoppered bottles, and accessing the gas phase from mylar bags in a quantitative fashion is challenging.

Measurement of radiolabeled dissolved inorganic carbon requires that all incubation material be transferred to an Erlenmeyer flask equipped with a scintilation vial; sediment grains are commonly trapped in the seals of mylar bags, complicating this transfer. For these reasons, monodeuterated methane addition and subsequent aqueous measurement offered a useful tool for this challenging experimental set-up.

Once the incubations were prepared, they were transported to the laboratory of Dr. William

Berelson at the University of Southern California and placed in a walk-in cold room (4 °C).

Incubations for pressurized treatment were inserted into a stainless steel, custom-built pressure chamber with 3-cm thick walls and pressure valves rated to 40 MPa, and hydraulic fluid was pumped into the sealed chamber using a Star Hydraulics P1A-250 hand pump. The pressure was maintained at 9.0 MPa during the course of the 38-day experiment, with daily adjustments to account for thermal compression effects. At the conclusion of the experiment, mylar bags were removed from the chamber, checked for leaks (none were observed, as the bags were still inflated, the seal was still gas-tight, and no hydraulic fluid was detected in the interior of the mylar bags)

and sampled for D/H ratio measurement.

2.2. Analytical Procedures

*2.2.1. Rate Measurements Derived from CH₃D Addition*

At designated sampling times, 1 mL of medium / water was collected from cultures or incubations in an anaerobic chamber with a sterile syringe. A constant volume was maintained by adding 1 mL of sterile media immediately after sampling; this media was previously equilibrated with gaseous headspace specific to each experiment. Sampled liquid was then pushed through a

0.22 μm Durapore filter (EMD Millipore, Temecula, CA) and into a 1-mL GC vial. A LGR DLT-

100 liquid water isotope analyzer (LWIA, Los Gatos Research, Mountain View, CA) was used to determine the D/H ratio of each sample. The LWIA uses off-axis integrated-cavity output spectroscopy to measure isotopically specific absorption patterns and determine simultaneous D/H

and $^{18}O/^{16}O$ ratios with high precision and robust mechanics (Lis et al., 2008). Such instruments have been used for a range of studies, including hydrological analysis (Robson and Webb, 2016), mine waste management (Huang et al., 2015), and microbial metabolism (Dawson et al., 2015).

In this study, an injection volume of 700 nL at 1000 nL/s was used, with four intra- injection flush strokes and a flush time of 60 s between injections. Four rounds of ten injections per sample were performed in order to avoid memory effects; only the latter five injections were used in subsequent calculations. Sample runs were limited to ~250 injections in order to minimize salt precipitation, and each analysis included an appropriate blank (i.e., autoclaved media for the cultures, or filter sterilized bottom water used during incubation set-up in the case of sediment and carbonate incubations) and two standards of known isotopic ratios (Deep Blue: $\delta D = 0.5‰$, and

CIT: $\delta D = -73.4‰$). Data was removed if instrumental temperature or pressure parameters were observed to fall outside of optimal instrument specifications (0.76% of all analyses), corresponding to an internal temperature change of more than 0.3 °C per hour or rising pressure within the measurement cell during the analysis.

To calculate methane consumption rates, D/H ratios from the LWIA were first normalized to the Vienna standard mean ocean water (VSMOW) scale using a two-point calibration from the water standards and a linear interpolation (e.g., Dawson et al., 2015). To minimize the effects of instrumental drift, standards were re-measured between rounds of sample analysis (maximum of

40 injections) and new scaling factors were implemented. The number of total hydrogen atoms (H

and D) present at the initial time point was calculated using the experiment's overall water volume, as in equation 1:

$$\frac{vol(L)}{1} * \frac{55.5 \; mol \; water}{L} * \frac{6.022 \; x \; 10^{23} \; molecules \; water}{mol \; water} * \frac{2 \; hydrogen \; atoms}{molecule \; water} = hyd. \, atoms \; in \; inc._{T1}$$

The number of D atoms newly present in the experiment's aqueous phase between time points $T_1$

and $T_2$ was determined using the normalized D/H values, averaging across the latter five injections of the four distinct injection rounds; see equations 2.1-2.3:

$$\left[\left(\frac{D}{H}\right)_{T2} * (hyd. \, atoms \; in \; inc.)_{T2}\right] - \left[\left(\frac{D}{H}\right)_{T1} * (hyd. \, atoms \; in \; inc.)_{T1}\right] = new \; D \; atoms \; in \; inc.$$

$$(hyd. \, atoms \; in \; inc.)_{T2} \approx (hyd. \, atoms \; in \; inc.)_{T1}$$

$$\left[\left(\frac{D}{H}\right)_{T2} - \left(\frac{D}{H}\right)_{T1}\right] * (hyd. \, atoms \; in \; inc.)_{T1} = new \; D \; atoms \; in \; inc. = D_{new}$$

$D_{new}$ was multiplied by four given the 1:3 D:H stoichiometry of the $CH_3D$ substrate to derive the maximum number of methane molecules consumed catabolically through initial C-X bond activation (equation 3).

$$D_{new} * 4 = maximum \; number \; of \; methane \; molecules \; consumed = C$$

The scaling factor of four was used in the context of methane activation – the initial mobilization through conversion to a methyl group – to calculate the maximum number of methane molecules that could be consumed but not necessarily fully oxidized. This represents an end-member case that may not be appropriate for all subsequent processing as hydrogen/deuterium atoms are incorporated into biomass or exchanged. Caveats and potential interpretations of the absolute numbers that result from these calculations are discussed below, but we stress that with consistent implementation of scaling factors from comparisons between monodeuterated and radiolabel methods, rates derived from $C$ and downstream parameters are valid and useful.

$C$ was corrected based on the fraction of incubation methane headspace composed of

CH$_3$D, yielding $C_{corr}$, as shown in equation 4:

$$\frac{C}{fraction\ of\ methane\ headspace\ that\ is\ CH3D} = C_{corr}$$

By dividing $C_{corr}$ by the incubation time and volume, a maximum rate of methane consumption is determined (equation 5.1-5.2):

$$C_{corr} * \frac{mol}{6.022\ x\ 10^{23}\ molecules} * \frac{10^9\ nmol}{mol} * \frac{1}{incubation\ time\ (d)} * \frac{1}{incubation\ vol\ (cm^3)} = R_{CH3D}$$

$$R_{CH3D} = Maximum\ rate\ of\ methane\ consumption\ in\ \frac{nmol}{cm^3\ d}$$

*2.2.2. Rate Measurements Derived from $^{14}CH_4$ Addition*

Methane oxidation rates using radiolabeled methane substrate were measured as described in detail by Treude et al. (2005) and Treude and Ziebis (2010). Radiolabeled methane ($^{14}$CH$_4$

dissolved in seawater, corresponding to an activity of 13 kBq for culture experiments and 52 kBq in sediment and carbonate samples) was injected into each sample container, and samples were incubated at the appropriate temperatures for the designated amount of time (see above). To stop microbial activity and begin analysis, 2.5 ml of 2.5% NaOH was injected. Sample headspace was flowed through a $Cu^{2+}$ oxide-filled 850 °C quartz tube furnace, combusting unreacted $^{14}CH_4$ to

$^{14}CO_2$. This $^{14}CO_2$ was collected in two scintillation vials (23 ml volume) pre-filled with 7 ml phenylethylamine and 1 ml 2-methoxyethanol, to which 10 ml of scintillation cocktail (Ultima

Gold XR, PerkinElmer) was added. After a 24-hour incubation period, radioactivity from $^{14}CO_2$

was measured by scintillation counting (Beckman Coulter LS 6500 Multi-Purpose Scintillation

Counter, 10 minute analysis per sample).

Labeled $^{14}C$-inorganic carbon produced during the incubation was quantified as follows.

The entire volume of each incubation sample was transferred into a 250-ml Erlenmeyer flask along with 1 drop of antifoam and 5 ml of 6M HCl. The flask was immediately stoppered and sealed with two clamps and parafilm wrapping to prevent gas escape, and placed on a shaking table (60

rpm, room temperature, 24 hours). To collect $^{14}CO_2$ generated by the acidification process, a 7-ml scintillation vial was pre-filled with 1 ml of 2.5% NaOH and 1 ml of phenylethylamine and suspended from the rubber stopper inside the flask. After the shaking / acidification step, 5 ml of scintillation cocktail was added, and the vial was measured by scintillation counting after 24 hours.

This method has been demonstrated to recover 98% of $^{14}CO_2$ on average (Treude et al., 2003).

Finally, sterilized control samples (#10, see Table S1) were set aside after $^{14}CH_4$ addition for gas chromatography to determine the initial concentration of methane gas. 400 μl of headspace was injected into a gas chromatograph (Shimadzu GC-2014), equipped with a packed stainless steel Supelco Custom Column (50/50 mixture, 80/100 Porapak N support, 80/100 Porapak Q

column, 6 ft x 1/8 in) and a flame ionization detector. The carrier gas was helium at a flow rate of

30 ml min$^{-1}$, and the column temperature was 60 °C. Results were scaled based on comparison with standards of known methane concentrations (10 and 100 ppm; Matheson Tri-Gas, Twinsburg,

OH). The rate of methane oxidation was determined by equation 6:

$$Methane\ Oxidation = \frac{^{14}CO_2 \bullet CH_4}{(^{14}CH_4 + {}^{14}CO_2) \bullet v \bullet t}$$

in which $^{14}CH_4$ is the combusted unreacted radiolabeled methane, $^{14}CO_2$ represents the quantity of acidified oxidation product, $CH_4$ signifies the initial quantity of methane in the experiment, v is the volume of sediment or carbonate rock, and t is the time over which the incubation was active.

*2.2.3. Isotopic Analysis of Methane in the Headspace*

The methane headspace was analyzed via $^1$H-NMR spectroscopy using a Varian 400 MHz

Spectrometer with a broadband auto-tune OneProbe. 300 µl of headspace was passed through

$CDCl_3$ with a fine needle to absorb the methane. $^1$H-NMR spectra were acquired at 298 K without spinning, using a repetition rate of 10 s to ensure reliable quantification. The spectra were simulated with the iNMR 4.1.7 software for the determination of the fractional abundances of the

$^{12}CH_4$, $^{12}CH_3D$, $^{13}CH_4$ and $^{13}CH_3D$ isotopologs.

**3. Results and Discussion**

3.1. Comparison of $CH_3D$ and $^{14}CH_4$ Rate Measurements in Aerobic Methanotroph Cultures

D/H ratios were acquired and corresponding $C_{corr}$ values were calculated at eight points during the *M. trichosporium* growth curve and seven points of the *M. sedimenti* growth curve.

Three measurements of $^{14}C$ distributions were acquired for each strain, targeting exponential and stationary phases (Fig. 1). The Type II alphaproteobacterial methanotroph *M. trichosporium*

exhibited methane consumption rates more than an order of magnitude greater than those of *M.*

*sedimenti* (gammaproteobacterial Type I methanotroph), yet the scaling factor relating the $CH_3D$- and $^{14}CH_4$-derived rates was remarkably consistent in both cases. Scaling factors were calculated for both exponential growth and stationary phase, using data points from both $CH_3D$ and $^{14}CH_4$

experiments. For example, the *M. trichosporium* rate value calculated from $CH_3D$ experimental treatment point (47.5 hours, 4.16 x $10^4$ nmol methane consumed) was compared with the rate determined from $^{14}CH_4$ experimental treatment point (47.5 hours, 2.77 x $10^4$ nmol methane consumed), yielding a scaling factor of 1.5 for exponential phase growth. Similarly, data from (140

h, 5.27 x $10^4$ nmol, $CH_3D$) and (166.5 h, 4.24 x $10^4$ nmol, $^{14}CH_4$) were used for *M. trichosporium*'s stationary phase scaling factor. Equivalent values were determined for *M. sedimenti* using the following data points: (140 h, 7.07 x $10^3$ nmol, $CH_3D$) and (102 h, 3.43 x $10^3$ nmol, $^{14}CH_4$) for the exponential growth phase, and (476 h, 7.53 x $10^3$ nmol, $CH_3D$) and (432 h, 4.30 x $10^3$ nmol,

$^{14}CH_4$) for stationary phase.

In this way, the ratio of methane consumption rates derived from the $CH_3D$ method (using equations 1-5) and the $^{14}CH_4$ method (using equation 6) can be compared. This value is hereafter referred to as the "D:$^{14}C$ tracer ratio". This ratio can be used to evaluate the consistency of the monodeuterated methane method compared with the well-established $^{14}CH_4$ approach, and as an investigatory tool in catabolic / anabolic processing of methane (see "Understanding the D:$^{14}C$

Tracer Ratio", below).

D:$^{14}C$ tracer ratio values were calculated for aerobic methanotroph cultures using the data specified above and are shown in Table 2; their consistency is a promising indicator of the utility of the monodeuterated methane approach for ground-truthed rate measurements. By dividing rates derived from D/H values by 1.5, a reliable estimate of full-oxidation methanotrophy – that is, the complete biological oxidation of methane to carbon dioxide – can be assessed.

3.2. Comparison of $CH_3D$ and $^{14}CH_4$ Rate Measurements in Environmental Methane Seep Samples

Methane consumption rates under oxic microcosm incubation conditions, derived from both $CH_3D$ and $^{14}CH_4$ measurements, are provided for all five sample types (active sediment, low- activity sediment, active porous carbonate, active massive carbonate, and low-activity massive carbonate) in Fig. 2a. The corresponding values for anoxic conditions are shown in Fig. 2b. Values were calculated from data collected after 4 days of incubation for oxic samples and after 8 days of incubation for anoxic samples.

The D:$^{14}$C tracer ratio for the oxic incubations was 1.66 +/- 0.02 SE and 1.99 +/- 0.04 SE

for anoxic conditions (Table 2). These relatively consistent values across physical substrate type (sediment and carbonates of varying lithology) and collection site activity level (active and low- activity) suggest an underlying metabolic basis of these D:$^{14}$C tracer ratio that is unperturbed by physicochemical factors or relative activity levels.

To determine the minimum number of activated $CH_3D$ molecules needed for analytical detection, we assessed the length of time required to measure a differentiable D/H ratio.

Measurements were acquired at multiple time points between days 20 and 22 of a triplicate set of

A.Sed-5128 incubations. A resolvable signal of an enhanced D/H ratio was defined as data points with non-overlapping confidence intervals, representing a 95% statistical probability that D/H

ratios were increased. Such differentiation seen at the 20-hour sampling time for two replicates and the 26-hour sampling time for the other one (Fig. S1). Using the rate determined by the first 20

days as a baseline, this translates to a resolution of 4.5-6.2 μmol of fully oxidized methane based on the D:$^{14}$C tracer ratio of 2.05 (Table 2).

3.3. Understanding the D:$^{14}$C Tracer Ratio

The $CH_3D$ and $^{14}CH_4$ approaches quantify distinct aspects of methanotrophy; that is, methane activation or complete conversion to $CO_2$, respectively. The $^{14}CH_4$ technique quantifies the amount of $^{14}C$ – initially supplied as methane – that is fully oxidized and persists as soluble species ($HCO_3^-$) or acid-labile precipitation products ($CaCO_3$). The $CH_3D$ protocol, on the other hand, reports the extent to which methane-derived hydrogen atoms are detected in water. Because methane is an inert molecule, abiotic exchange between methane- and water-associated hydrogen atoms is not expected. Indeed, D/H ratios in killed control experiments remained stable (e.g., exhibiting a value of $1.40 \times 10^{-4}$ +/- $3.1 \times 10^{-8}$ SE at $T_{0d}$ and $1.40 \times 10^{-4}$ +/- $2.9 \times 10^{-8}$ SE at $T_{140d}$

during experimentation with *M. trichosporium*, data incorporated into Fig. 1a). The activation of methane thereby indicates enzymatic functionalization, but the ultimate fate of each hydrogen atom during methane oxidation is unclear.

[revised manuscript text omitted]

tracer ratios were similar for both cultured organisms despite their distinct metabolic pathways; a similar phenomenon of consistent carbon conversion efficiency was recently observed among distinct aerobic methanotroph communities in English riverbeds (Trimmer et al., 2015).

The oxic incubations of methane seep sediment produced a D:$^{14}$C tracer ratio of 1.66 +/-

0.02 SE. Given that the known modes of biological methane oxidation – type I and type II aerobic methanotrophy and reverse methanogenesis anaerobic methanotrophy – bound this observed value, it appears likely that the oxic sediment incubations supported a mixture of both aerobic and anaerobic methane oxidation processes. Aerobic methane oxidation likely dominated, based on the

$\sim 7 \times 10^4$ Pa partial pressure of $O_2$ and the proximity of the D:$^{14}$C tracer ratio to that of the aerobic methanotrophic cultures, but anoxic niches likely remained or developed in the incubation bottles.

3.4. Specialized Application of the Monodeuterated Methane Approach: Anaerobic Methanotrophy at Pressure

To demonstrate the utility of the CH$_3$D rate measurement approach in addressing experimentally relevant questions, particularly in nontraditional empirical contexts, we sought to evaluate the influence of *in situ* pressure on methanotrophic rates of Hydrate Ridge seep sediment microbial communities. Material collected for microbiological studies of AOM is frequently obtained from marine settings of various depths that are subjected to distinct and substantial pressure regimes (Ruff et al., 2015). Pressure is not always rigorously incorporated into microcosm experiments, though evidence suggests it can be an important determinant of methanotrophic rates (Bowles et al., 2011; Nauhaus et al., 2005; Zhang et al., 2010). In addition, some procedural aspects of the $^{14}$CH$_4$ protocol, including headspace sampling and full-volume transfer, are not established for use with mylar bags, making the monodeuterated methane approach an appealing alternative in this context.

Parallel seep sediment incubations were subjected to 0.1 MPa (atmospheric pressure) and

9.0 MPa (equivalent to ~900 m depth). δD values derived from heightened D/H ratios attributable to methane consumption, are shown in Fig. 5. A significant increase in methane consumption was observed in both live conditions at heightened pressure, corresponding to sediment incubated with glycine (samples 1a and 1b) and ammonium chloride (samples 2a and 2b). Controls lacking CH$_3$D

(samples 3a and 3b) and biological activity (samples 4a and 4b) showed no increase in δD (see

Table S2 for sample set-up details). The simulation of *in situ* Hydrate Ridge pressures led to a

79.5% (+/- 6.5 SE) increase in relative methane oxidizing rates. Incubation with 500 μM glycine rather than ammonium at high and low pressures resulted in small but consistent rate increases of

12% +/- 4.1% SE, potentially reflecting the energetic and biosynthetic distinction between exogenous amino acids and unprocessed fixed nitrogen.

Previous reports have found a wide range of different pressure-related effects. In a sulfate- coupled AOM bioreactor, pressures were varied from 1 to 8 MPa and sulfide production approximately tripled, demonstrating Michaelis-Menten style kinetics with an apparent $K_m$ of 37

mM (Zhang et al., 2010). Methane partial pressures of 1.1 MPa led to a 5x increase in sulfate reduction rates relative to ambient atmospheric pressure with Hydrate Ridge sediments demonstrating methane-dependent sulfate reduction (Nauhaus et al., 2002). With methane seep sediment from the Japan Trench, however, methane-driven sulfate reduction rates did not correlate with changing pressure (Vossmeyer et al., 2012). Nauhaus et al. (2005) suggested that the pressure- induced rate increases are due more to heightened methane solubility and bioavailability rather than physiological effects or biomolecular re-ordering. Bowles et al. (2011) presented a very different perspective by showing a six- to ten-fold AOM rate increase at 10 MPa when methane concentrations were held constant. Deconvolving these two influences and how they depend on community composition or physicochemical parameters is feasible with pressure chamber experiments utilizing monodeuterated methane. Understanding the relative contributions of environmental and physiological effects to methane oxidation will help constrain methane fluxes across a larger envelope of the planet's methanotrophically active zones.

3.5. Using Monodeuterated Methane in Experimental Investigations

Based on $^{14}CH_4$ ground-truthing experiments with aerobic methanotrophic cultures, oxic seep sediment, and anoxic seep sediment, as well as the proof-of-concept pressurized experiments, we believe that the monodeuterated methane approach to methane oxidation rate measurement is a useful addition to the biogeochemist's tool set. Compared with radiolabel approaches ($^{14}CH_4$, $^3H$-

$CH_4$, $^{35}SO_4^{2-}$), the method requires less safety-oriented planning, and is logistically simpler, more affordable, and may be less susceptible to hydrogen-associated isotope fractionation effects (relative to $^3H$). Our results also suggest that the monodeuterated methane technique appears to be a more precise method based on standard error calculations (Figs. 1, 2). Direct comparisons of environmental incubations are complicated by the microheterogeniety of seep settings (Barry et al.,

1996; Lloyd et al., 2010), as well as the fact that different aliquots of the same initial material were used in our experiments. However, analysis of culture-based experiments reveals that standard errors from $R_{CH3D}$ values were 20% those derived from $^{14}CH_4$-based values, making the monodeuterated method five times more precise.

Because the monodeuterated methane method focuses on methane-bound hydrogen atoms, it offers different, complementary information about methanotrophic systems than carbon-based techniques like methane or bicarbonate quantification. While this distinction complicates the interpretation of isolated D/H ratios, it can offer an additional dimension of information for analysis of methane-derived intermediates in relevant metabolisms. Given these caveats, we recommend three use cases for monodeuterated methane in methane oxidation rate measurement applications.

1)    First, the approach can be employed in a strictly comparative context using analogous inoculum exposed to a range of different conditions, as demonstrated with the pressure- based sediment incubations presented above. Evaluating the effect of different

| 523 | | conditions such as temperature ranges, chemical concentrations, or energetic landscapes |
| 524 | | on seep sediment methane oxidizing rates would all be promising applications. |
| 525 | | Comparative analysis of AOM rates at different seep sites would also be useful, |
| 526 | | provided anaerobic or aerobic methanotrophic processes could be isolated. |
| 527 | 2) | Second, by performing side-by-side monodeuterated methane and radiocarbon tests, a |
| 528 | | sample-specific $D:^{14}C$ tracer ratio can be determined, and absolute rates of full methane |
| 529 | | oxidation can then be inferred in subsequent experiments based exclusively on D/H |
| 530 | | ratios. Conducting such paired studies under additional environmental or lab-based |
| 531 | | conditions would help clarify the universality of the ratios presented here and would |
| 532 | | likely reveal additional questions of metabolic dynamics in a range of experimental |
| 533 | | systems. We also encourage side-by-side comparisons with other rate measurement |
| 534 | | approaches, including $^{3}H$-$CH_4$ radiotracer and methane concentration assessments, to |
| 535 | | develop additional pairwise conversion factors and better constrain carbon and hydrogen |
| 536 | | metabolism in methane-based biological reactions. |
| 537 | 3) | Finally, the use of monodeuterated methane as an analytical tool, alongside additional |
| 538 | | methods such as carbon- or sulfur-tracking procedures, would enable a multi- |
| 539 | | dimensional examination of anabolic and catabolic processes in methane-based |
| 540 | | metabolisms. In particular, the $D:^{14}C$ tracer ratios presented here reveal intriguing and |
| 541 | | seemingly systematic relationships between carbon and hydrogen anabolic and catabolic |
| 542 | | partitioning across distinct physiologies, yet an underlying theoretical framework |
| 543 | | regarding the fate of methane-bound hydrogen atoms remains outstanding. In anaerobic |
| 544 | | methanotrophic systems, back-reaction rates and equilibrium constants could be |
| 545 | | evaluated by a) including a $^{13}CO_2$ signal in the water and measuring $^{13}CH_4$, and/or b) |

utilizing multiply deuterated methane as initial headspace and measuring all possible isotopologues via NMR or high resolution mass spectrometry. For aerobic methanotrophs, evaluating D:$^{14}$C tracer ratios under more clearly defined growth and maintenance phases would elucidate distinct values associated with catabolic, RuMP, and serine pathways, enabling future use of that parameter as an arbiter of relative anabolic and catabolic activity. Furthermore, additional environmental variables can be tested to gain insight into distinct redox pathways and dynamics of reversibility. For example, AOM under lower sulfate concentrations might be expected to generate higher

D:$^{14}$C tracer ratios (Yoshinaga et al., 2014), and this parameter could be further developed as a measure of microbially mediated isotopic equilibration.

**4. Conclusion**

The ability to accurately measure methane consumption and oxidation rates – both comparatively and in absolute values – is an important component of methanotrophic studies. Such measurements frequently depend on radiotracers or measurements of chemical species that are related to, but not directly indicative of, methane metabolism. The monodeuterated methane technique presented here represents a novel approach to methane oxidation rate measurements, notable for its logistical and analytical ease (particularly in ship-board applications), as well as the added dimension provided by H-based, rather than C-based, information. We have demonstrated that the D/H ratio is a reliable proxy for methane oxidation activity: in several applications, the value is directly proportional to methane oxidation rates as measured in absolute terms by the well- established $^{14}$CH$_4$ method. The value of the proportionality constant differs based on the experimental system, likely dictated by environmental variables and the relative proportions of aerobic and anaerobic methanotrophic metabolisms, though additional experiments to determine the nature of the putative mixing line are needed.

Methane biogeochemistry is a dynamic field of study with implications for carbon cycling, microbial ecology, and climate dynamics, though experimental challenges have slowed our understanding of methane-based biological reactions. With the $CH_3D$ approach as an added tool in the arsenal of rate-based examinations, a broader understanding of the intricacies of methane metabolism, as well as its role in environmental and anthropogenic systems, is within reach.

**5. Acknowledgements**

We thank the Captains, Crew, *Alvin* group, *Jason* group, and Science party members from

*RV Atlantis* legs AT-15-68, and AT-18-10. Water analyzer measurements were conducted in the laboratory of Alex Sessions at the California Institute of Technology with technical support from

Lichun Zhang. We are indebted to William Berelson at the University of Southern California and

Nick Rollins for use of their pressure chambers and assistance with the incubation experiments.

We thank Alex Sessions, Woodward Fischer, Dianne Newman, Tori Hoehler, Amy Rosenzweig,

Daniel Stolper, and Linda Reynard for helpful conversations during the preparation of the manuscript. This study was funded by grants from the U.S. Department of Energy, Office of

Science, Office of Biological and Environmental Research (DE-SC001057), the NASA

Astrobiology Institute (Award # NNA13AA92A) and support from the Gordon and Betty Moore

Foundation through grant GBMF3780 (to VJO). JJM was supported by a National Energy

Technology Laboratory Methane Hydrate Research Fellowship funded by the National Research

Council of the National Academies. This research used resources of the Oak Ridge Leadership

Computing Facility. Oak Ridge National Laboratory is supported by the Office of Science of the

U.S. Department of Energy.

[revised manuscript text omitted]

Fig. 1: Amount of methane consumed over time for cultures of a) the type II methanotroph *M. trichosporium* and b) the type I methanotroph *M. sedimenti* using $C_{corr}$ (values derived from the $CH_3D$ method, shown with circles) and the $^{14}CH_4$ method (diamonds), calculated as discussed in the text. $^{14}CH_4$-derived data conveys values of methane consumption and full oxidation, while $CH_3D$-derived data provides a measure of methane consumption. Error bars show standard errors for three biological replicates, with the exception of the $^{14}CH_4$–derived killed control (n=1). Obscured data points exhibited values between -60 and 110 nmol for a) and 0 and 60 nmol for b).

Fig. 2: Methanotrophy in a) oxic and b) anoxic incubations of active and inactive seep sediment and carbonate rocks (n=3 in all cases). Values compare methane consumption / full oxidation rates derived from $^{14}CH_4$ measurements (blue) and methane consumption rates derived from the $CH_3D$ approach (green, $R_{CH3D}$ values). Standard error bars are provided.

Fig. 3: A schematic diagram demonstrating the potential fate of methane-associated hydrogen atoms in the "reverse methanogenesis" pathway. Hydrogen atoms are distinguished by color and superscript number, and potential exchanges with inter- and intra-cellular water are shown.

Potentially detectable methane-derived hydrogen atoms (4, occurring throughout the oxidation pathway) and carbon atoms (1, requiring full oxidation) are highlighted in orange and purple boxes, respectively. Shorter "backflux" arrows reflect the observation that all enzymes (Thauer,

2008) and the entire pathway (Holler et al., 2011) have been shown to be reversible. For figure simplicity, isotopically distinct backflux products and cofactor involvement in backflux reactions are not shown. The extended dashed line represents the cell membrane.

Fig. 4: A schematic diagram demonstrating the potential fate of methane-associated hydrogen atoms in the aerobic methanotrophy pathways. Hydrogen atoms are distinguished by color and superscript number; asterisks represent location-specific ambiguity. Potentially detectable methane-derived hydrogen atoms and carbon atoms are highlighted in orange and purple boxes, respectively. Mmo enzymes are not believed to perform reversible reactions.

Fig. 5: Pressure experiment results showing water $\delta D$ values with standard error bars of seep sediment samples following 38-day incubations with $CH_3D$ at 9.0 MPa (brown bars, "b" samples)

or 0.1 MPa (pink bars, "a" samples). Additional details on sample treatments can be found in Table

S2.

**Figure 1**

a) Methanotrophy in *M. trichosporium* Cultures b) Methanotrophy in *M. sedimenti* Cultures

Legend (both panels):
- CH3D experimental treatment
- CH3D killed control
- CH3D abiotic control
- CH3D oxygen free control
- CH4 live control
- 14CH4 experimental treatment
- 14CH4 killed control

Figure 2

a)

[Figure]

Methanotrophy Under Oxic Conditions

Methane Consumption Rate (nmol / cm³ d)

$^{14}CH_4$
$CH_3D$

b)

[Figure]

Methanotrophy Under Anoxic Conditions

Methane Consumption Rate (nmol / cm³ d)

**Figure 3**

**"Reverse Methanogenesis" Pathway**

[Figure]

**Figure 4**

**Aerobic Methanotrophy Pathway**

[Figure]

Figure 5

[Figure]

Figure S1

[Figure]

Supplementary Material for ""**Monodeuterated Methane: An Isotopic Probe to Measure Biological Methane Metabolism Rates and Track Catabolic Exchange Reactions**"

Table S1: Conditions for the aerobic methanotrophy experiments. All sample types were set up in triplicate; pressures shown as partial pressures.

| Sample Condition | p(CH$_3$D) | p(CH$_4$) | $^{14}$CH$_4$ | p(O$_2$) | p(Ar) | Inoculum Introduced | Killed Cells |
|---|---|---|---|---|---|---|---|
| 1 | 0.1 MPa | | | 1 x 10$^5$ Pa | | 10% v/v | |
| 2 | 0.1 MPa | | | 1 x 10$^5$ Pa | | 10% v/v | Yes |
| 3 | 0.1 MPa | | | 1 x 10$^5$ Pa | | | |
| 4 | 0.1 MPa | | | | 1 x 10$^5$ Pa | 10% v/v | |
| 5 | | 0.1 MPa | | 1 x 10$^5$ Pa | | 10% v/v | |
| 6 | 0.1 MPa | | 13 kBq (T1) | 1 x 10$^5$ Pa | | 10% v/v | |
| 7 | 0.1 MPa | | 13 kBq (T2) | 1 x 10$^5$ Pa | | 10% v/v | |
| 8 | 0.1 MPa | | 13 kBq (T3) | 1 x 10$^5$ Pa | | 10% v/v | |
| 9 | 0.1 MPa | | | 1 x 10$^5$ Pa | | 10% v/v | |
| 10 | 0.1 MPa | | 13 kBq (T3) | 1 x 10$^5$ Pa | | 10% v/v | Yes |

Table S2: The experimental set-up for methane seep sediment pressurized rate measurement incubations. The samples ran for 38 days at 4 °C, and each sample was contained in a sealed Mylar bag. Pressure values indicate absolute pressure exerted on the incubated Mylar bags.

| Sample # | Sediment | Nitrogen Source | Methane Source | Pressure (MPa) |
|---|---|---|---|---|
| 1a | 50 mL | 500 uM Glycine | 40 mL CH$_3$D | 0.1 |
| 2a | 50 mL | 500 uM NH$_4$Cl | 40 mL CH$_3$D | 0.1 |
| 3a | 50 mL | 500 uM NH$_4$Cl | 40 mL CH$_4$ | 0.1 |
| 4a | 50 mL, killed control | 500 uM Glycine | 40 mL CH$_3$D | 0.1 |
| 1b | 50 mL | 500 uM Glycine | 40 mL CH$_3$D | 9.0 |
| 2b | 50 mL | 500 uM NH$_4$Cl | 40 mL CH$_3$D | 9.0 |
| 3b | 50 mL | 500 uM NH$_4$Cl | 40 mL CH$_4$ | 9.0 |
| 4b | 50 mL, killed control | 500 uM Glycine | 40 mL CH$_3$D | 9.0 |

Fig. S1: To assess the empirical resolving power of the D/H measurement technique, we determined the time points showing non-overlapping confidence intervals for triplicate incubations of A.Sed-5128. Distinct signals were seen at the 20-hour time point for replicate A (teal diamonds) and the 26-hour time point for replicates B (pink diamonds) and C (green diamonds).

[Figure]

---

## Author Comment (AC2) · 18 Aug 2016

**Reviewer 1**

Marlow and coworkers tested D-labeling of methane as an alternative tracer based method to determine methane oxidation rates in aerobic methanotrophic cultures, and in oxic and anoxic (AOM-active) sediments. D-labeled methane was added to a sample and the change of water deuterium isotopic composition was measured using Cavity-Ringdown Laser Absorption Spectroscopy (Los Gatos). Values were used to determine rates of methane oxidation. Alternatively the 14C-methane method was used and rates were compared. The authors found generally higher rates with the D-methane compared to 14C-methane tracer approach. As reasons for this, although not really clearly pointed, two mechanisms were discussed. Most aerobic methanotrophs use primarily methane as carbon source - hence 14C-methane is partly transferred into organic biomass and not into CO2. Methane-H is instead almost fully transferred into water, hence seems to be promising for an accurate measurement of aerobic methanotrophy. In anaerobic methanotrophy back reaction that cause isotopic exchange were identified, which would lead to rather overestimated real rates. In general reported values have a good precision, meaning a low standard deviation – however it has not been tested if the rates are comparable to chemical measurements and as such lacks a trueness/ accuracy validation. The manuscript is full of slang, and often misses accurate terms, and lacks structure. Due to this in particular the discussion is not easy to follow. Furthermore section titles are not informative. This would need to be improved.

Furthermore one has to criticize that the main, new method is insufficiently described. Why is there no formula? Instead the new approach is only described in text. From this I at least think that the rate determination is slightly incorrect. As far as I can anticipate the authors use ratios instead of (required) fractions in their calculation. Please also refer to the labeling percentage you used. Below are my further comments, which are not complete – because sometimes I could simply not follow the discussions!

Specific Comments

**1.** The abstract is not really productive, here a suggestion on how to improve it: "Biological methane oxidation is a globally relevant sink of methane and it proceeds in aerobic and anaerobic pathways. However measuring rates of methane oxidation in natural samples remains challenging. Here we present a new approach for measuring rates of microbial methane oxidation that bases on the addition of labeled monodeuterated methane (CH3D) to a sample, and quantification of the label in water phase via isotope ratio mass spectrometry. As comparison we performed the well-established 14CH4 radiotracer approach the CH3D procedure. We provide measurement with cultures of I and type II aerobic methanotrophs and for sediment and carbonate rock samples incubated under anoxic and oxic incubation conditions. [results] [what is the different and why is it] We also employ this method to investigate the role of pressure on methane oxidation rates in anoxic seep sediment, revealing an 80% increase at the equivalent of ~900 m water depth (40 MPa). [Conclusion ; where and why should the new method be used]

We thank the reviewer for the proposed changes to the abstract, and have incorporated some of them; specifically, the description of the new method has been streamlined. However, we have maintained most of the abstract as previously composed (and modified it in accordance with

other reviewer comments) because it includes important context about the strategic advantages of the CH$_3$D approach and specifically outlines three use cases. We believe these components of the abstract are useful for readers to understand how the method is different and how it may be useful in additional research applications.

**2.** L47: "Often involving"? where is the real counter evidence/ or do you mean the Haroon Paper?

The use of this qualifier allows for the inclusion of many anaerobic methanotrophic metabolisms that may or may not be linked with sulfate reducing bacteria. Most of these "alternative" approaches are listed later in the paragraph (lines 59-62, including studies from Haroon et al., 2013; Ettwig et al., 2010; Beal et al., 2009; Nauhaus et al., 2005; and Sivan et al., 2014); while not specifically cited, the possibility of free-living ANME also remains unsettled (Lloyd et al., 2006).

**3.** L50: I would cite the Mc Glynn and Wegener nature studies on the mechanism for AOM, they seem to be the most promising approaches, also considering the new Scheller study

These additional citation recommendations have now been incorporated into the text.

**4.** Same Paragraph: As later aerobic methanotrophy is discussed it should also be introduced here

We thank the reviewer for pointing out this gap in our introductory section. The phylogenetic and metabolic background of aerobic methanotrophy is now summarized in lines 63-71, which read as follows: "Methane is oxidized aerobically by members of the classes *Gammaproteobacteria* (Type I) and *Alphaproteobacteria* (Type II); verrucomicrobial representatives were more recently found to perform aerobic methanotrophy under extremely acidic conditions (Dunfield et al., 2007; Op den Camp et al., 2009). Methane monooxygenase converts methane to methanol, which is further oxidized to formaldehyde; assimilatory pathways branching at this point can incorporate carbon into central metabolism through the ribulose monophosphate (RuMP) cycle (Type I methanotrophs) or the serine cycle (Type II). Remaining formaldehyde can undergo two additional oxidation reactions, being converted first to formate and ultimately to carbon dioxide (Hakemian and Rosenzweig, 2007)."

**5.** L56: The relevance in nature was discussed above, does not need to be repeated here

While it is true that methanotrophy's environmental importance was mentioned previously, the restatement here establishes the context for widespread interest in measuring rates of the process. Because this reminder is a short phrase and is part of a clause that unifies the previous two paragraphs, we trust that most readers will find the recapitulation useful.

**6.** L58: "AOM rate measurements have traditionally…" The performed measurement does not discriminate aerobic or anaerobic methane oxidation – I would rephrase it: The turnover of methane to carbon dioxide in environmental samples is often traced using stable or radiocarbon labeling approaches

We appreciate the reviewer's note on this point, and the sentence now reads as follows (lines 75-76): "The oxidation of methane in environmental samples has traditionally been studied using a handful of techniques." This broader phrasing not only encompasses aerobic and anaerobic metabolism, but also allows for methods that follow hydrogen or carbon atoms. (Specifying "methane to carbon dioxide" would neglect tritiated methane methods.)

**7.** L61: Stable isotope 13CH4 tracers is not correct: Methane labelled with 13C or something

This proposed modification has been incorporated into the text, which now reads as follows (line 78): "Methane labeled with $^{13}$C can be used…"

**8.** L62: but the natural presence of 13C in marine dissolved carbon requires accurate detection of the reactant and product concentration and isotopic compositions. (I think that is what you mean.)

To provide additional detail on the challenges of $^{13}$C based methods, the text now reads as follows (lines 79-81): "…but the presence of natural $^{13}$C in marine dissolved inorganic carbon pools requires long incubations as well as accurate measurements of concentrations and isotopic compositions of reactants and products (Pack et al., 2011)."

**9.** L66: "though low molarities make samples susceptible to exsolution…" what should this mean? " I think you wanted to elaborate on concentration/ diffusion profiles…but then you need to tell a full story…that it really not enough

We thank the reviewer for pointing this out and have now included more detail regarding the challenges of GC-based rate quantification. The sentence now reads as follows (lines 83-85): "…though low concentrations can hamper reproducibility and exacerbate background contamination issues, particularly in field-based settings (Magen et al., 2014)."

**10.** L68: "carbon movement into oxidized species" … Does not sound scientific – track the oxidation of methane to CO2

The phrase now reads as follows (85-86): "…to track the oxidation of methane-associated carbon to inorganic carbon species."

**11.** L69: Labeling with tritiated methane was introduced to track aerobic methane turnover in the water column. The shorter half-life time allows higher specific activity, and the procedures for the separation of reactant and product are less complicated and can thus be performed on sea.

To capture the precision requested by the reviewer, this passage now reads as follows (lines 87-90): "Labeling with tritiated methane was introduced for water column aerobic methane oxidation measurements due to its higher specific activity and the procedural advantages of working with a water-phase product rather than gaseous products (Bussmann et al., 2015; Valentine et al., 2001)."

**12.** L75: measurements of methane turnover remain

This phrase has been changed to the following (lines 94-95): "Despite the range of methods available, measurement of microbial methane utilization rates remains cumbersome…" We believe that "microbial methane utilization" is a more appropriate term than "methane turnover"; the former specifies any microbially mediated process that draws down methane and is end product-agnostic, while the latter does not distinguish biotic from abiotic processes and does not specify the end product(s).

**13.** L77: delete: "Nearly all of the aforementioned approaches are carbon-based; a hydrogen-based tracer offers an additional dimension to investigations of methane biochemical dynamics."

This suggestion was not incorporated into the text; we believe that the hydrogen atom tracer aspect of monodeuterated methane is an important point of distinction compared with most other approaches. This point is discussed at greater length later in the manuscript, and setting the stage for that discussion is an important aim of the introductory section.

**14.** L83: "that aqueous D/H values are consistently proportional to 14C-based rate measurements for given laboratory treatments tested in this study…" Of course it is the rates derived from the comparison of… D/H are consistently proportional to those measured. But better remove 82-92 : those are your results

This paragraph has been retained. We believe that by recapping the key points of the introduction and succinctly previewing the coming results and discussion, this section orients the reader for the remainder of the manuscript. In accordance with the reviewer's request, the first sentence now reads as follows (lines 101-104): "We demonstrate, through methanotrophic cell cultures and microcosm incubations of seafloor sediment and carbonate rock fragments, that methane consumption rates derived from aqueous D/H values are consistently proportional to $^{14}$C-based rate measurements for given laboratory treatments tested in this study."

Methods

**15.** 2.1.1. Aerobic Methanotroph Cultures it is : Experiments with cultured aerobic methanotrophic bacteria

We appreciate the proposed section and subsection title suggestions from multiple reviewers, and have made several adjustments. Please see below for the new hierarchy of section headings:

1. Introduction
2. Methods
    2.1. Experimental Set-Up
        2.1.1. Experiments with Aerobic Methanotroph Cultures
        2.1.2. Experiments with Environmental Samples: Methane Seep Sediments and Carbonates
        2.1.3. Experiments with Environmental Samples in Pressure Vessels
    2.2. Analytical Procedures
        2.2.1. Rate Measurements Derived from $CH_3D$ Addition
        2.2.2. Rate Measurements Derived from $^{14}CH_4$ Addition
        2.2.3. Isotopic Analysis of Methane in the Headspace
3. Results and Discussion

**16.** After several successful transfers – how is a transfer successful – how did you see that cells were growing

Cell proliferation, a marker of a successful transfer, was measured by optical density at 600 nm wavelength. This sentence has now been adjusted as follows (lines 139-140): "After several successful transfers (as determined by an increase in optical density, data not shown)…"

**17.** L116: Parallel incubations incorporated 14CH4 to allow for three time points of destructive sampling, and killed and radiolabel-free controls. In parallel incubation we tested the turnover of 14CH4… And then go on with a clear description as above

To clarify the set of treatments used for radiolabel rate studies, this sentence now reads as follows (lines 142-144): "Due to the destructive nature of the $^{14}CH_4$ method, three of these triplicate sets were used to measure methane oxidation at three distinct time points." Discussion of the control treatments is now at the end of section 2.1.1.

**18.** L121: Here you could have compared it with purely chemical measurements

We agree that pairwise comparisons of the new monodeuterated methane approach to all other methane-consumption rate measurement methods would be useful. However, such an endeavor was beyond the scope of this study, and would be supplemental to the primary aim of our work described here. The radiolabeled $^{14}CH_4$-based approach is commonly accepted as a sole method for methanotrophic rate quantification (e.g., Thomsen et al., 2001, *AEM*; Kallmeyer & Boetius, 2004, *AEM*; Orcutt et al., 2005, *GCA*; Treude et al., 2005, *GCA*; Niemann et al., 2006, *GCA*; Treude et al., 2007, *AEM*; Bowles et al., 2011, *GCA*; Holler et al., 2011, *ISME J*; Milucka et al., *Nature*, 2012; Ruff et al., *PLoS One*, 2013; Segarra et al., *GCA*, 2013; Marlow et al., 2014, *Nature Communications*; Ruff et al., *Frontiers in Microbiology*, 2016). Given this extensive heritage, we believe it is appropriate to treat the radiolabeled $^{14}CH_4$-based approach as a suitable "standard" method against which new protocols should be compared.

In addition, we point the reviewer to studies that have connected "purely chemical measurements" – which we interpret to mean dissolved or headspace methane concentrations measured over time – with radiolabeled $^{14}CH_4$-based methods. Our link between $CH_3D$ and $^{14}CH_4$ approaches can then be connected (albeit indirectly) to chemical measurements as established in these previous publications. Most notably, Treude et al., (2003) reveal a consistent relationship between short-term radiotracer rate measurements and longer methane concentration measurements in Hydrate Ridge cores across multiple habitats.

Clarification of both of these points has been added to the first paragraph of the Methods section, which now begins as follows (lines 118-123): "To demonstrate the precision and reproducibility of the monodeuterated methane approach, it was tested alongside the well-established $^{14}CH_4$ radiotracer protocol. The use of $^{14}CH_4$ is an accepted standard procedure in studies of methane consumption quantification (e.g., Knittel and Boetius, 2009; Ruff et al., 2016; Segarra et al., 2013) and has been experimentally cross-referenced with methane concentration measurements (Treude et al., 2003) and other approaches including tritiated methane techniques (Pack et al., 2011; Mau et al., 2013)."

We have also added a passage in the discussion (lines 533-536) promoting more direct comparative study: "We also encourage side-by-side comparisons with other rate measurement approaches, including $^3H$-$CH_4$ radiotracer and methane concentration assessments, to develop additional pairwise conversion factors and better constrain carbon and hydrogen metabolism in methane-based biological reactions."

**19.** 2.1.2. Environmental Samples: Methane Seep Sediments and Carbonates Again that is not an informative headline: Measurement of methane turnover in seep sediments and seep carbonates

Please see our response to comment #15 above.

**20.** L137-46: Unnecessary

This section was retained, as the thorough description of sample labeling and origin offers important context for readers – many of whom may not be intimately familiar with seep systems – and prepares them for the data presented in the manuscript's tables and figures. Descriptions of carbonate rocks are helpful in demonstrating that the consistent rate-based patterns shown later are not dependent upon lithological characteristics but rather are likely linked to microbiological processes, a point that broadens the method's applicability to a wider range of sample types. (In addition, another reviewer requested an augmentation of this section.)

**21.** L192: Analytical procedures 2.2.1 Rename – that is not a CH3D rate measurement. Determination of rates based on deuterium oxide formation? Or something similar.

The reviewer's point is well taken, and based on this suggestion, section 2.2.1 was renamed to "Rate Measurements Derived from $CH_3D$ Addition." We prefer to keep $CH_3D$ in the heading to signal to the reader that the section that follows describes the generation of data from experiments with monodeuterated methane. In the same spirit, section 2.2.2 was renamed "Rate Measurements Derived from $^{14}CH_4$ Addition."

**22.** Bring also the D-Methane concentration first – or was this not done at all?

The concentration of methane that was composed of monodeuterated methane is now incorporated into the calculations shown throughout section 2.2.1 (specifically, equation 4).

**23.** Methods: I don't see a formula how D-values are transformed into rates – even though it a new method which is described. As a quick remark, to be mathematically correct and fully

comparable, mathematically you cannot use ratios – you need to use fractions instead
FD=R(DH)/(R(DH)+1) And please revise all your downstream calculations

We agree that a detailed formula showing readers how to convert from raw data coming off the water analyzer (e.g., D/H ratios) to rates of methane activation is important and appreciate the reviewer bringing this to our attention. The formula is now integrated into the text of section 2.2.1, which has been expanded to include the relevant details (found within lines 261-288).

**24.** How much of the methane was used up in total in your approach how much methane was added. Did you do any control measurement????

Relevant controls were incorporated into all experimental procedures described in this manuscript. In particular, aerobic methanotroph culture controls are detailed on lines 150-151 and Table S1; environmental sample controls are detailed on lines 171-174, 199-200; and pressure experiment controls are discussed on lines 213-216 (part of which was newly added) and shown in Table S2. Methane concentrations were not quantified; see response to comment #18 above.

**25.** L234: Name in "Labeled 14C-inorganic carbon or define it once- it was before at pH14, so only CO3–.

This suggestion has been incorporated into the text. The phrase (line 299) now reads as follows: "Labeled $^{14}$C-inorganic carbon produced during the incubation…"

**26.** L255: Your main new method is misplaced and only briefly described – you need to mention how you go from the spectra (which is the input of labeled methane) to the D2O

The description and calculation pertaining to the monodeuterated methane approach has been substantially expanded in response to this and other comments. Please see section 2.2.1 in the revised manuscript.

**27.** 3. Results and discussion 3.1. "Aerobic methanotrophic cultures" is not a title that can refer to a result – name it Comparison of rate measurements in aerobic methanotrophic cultures 3.2. Similar as above 3.2 needs a complete revision – I don't understand it Since 3.1 and 3.2 are only results one could make them results, 3.3. could be discussion

Please see our response to comment #15 above.

**28.** 3.3. It is not the "H:C tracer ratio" that needs understanding here, but the rates derived from 14CH4 and CDH3 incubations

We agree with the sentiment conveyed through this comment. However, because the most precise description of this ratio is wordy and unwieldy, and because we refer to it throughout the remainder of the manuscript, a shorthand phrase is needed. To accommodate the needs of concision and precision, we have changed this phrase from the "H:C tracer ratio" to the "D:$^{14}$C tracer ratio," which more accurately describes the origin of the data used to determine the ratio's

value. The passage defining this term reads as follows (lines 347-352): "In this way, the ratio of methane consumption rates derived from the $CH_3D$ method (using equations 1-5) and the $^{14}CH_4$ method (using equation 6) can be compared. This value is hereafter referred to as the "D:$^{14}C$ tracer ratio". This ratio can be used to evaluate the consistency of the monodeuterated methane method compared with the well-established $^{14}CH_4$ approach, and as an investigatory tool in catabolic / anabolic processing of methane (see "Understanding the D:$^{14}C$ Tracer Ratio", below)."

Regarding section and sub-section headings, please see our response to comment #15 above.

**29.** "The CH3D and 14CH4 approaches quantify distinct aspects of methanotrophy,"??? I thought you mention both quantify turnover of methane?

We have tried to clarify this further in the text. While both approaches measure the biological oxidation of methane, there is an important distinction between the $CH_3D$ and $^{14}CH_4$ methods. As described in the two sentences following this phrase (lines 380-384), "The $CH_3D$ and $^{14}CH_4$ approaches quantify distinct aspects of methanotrophy; that is, methane activation or complete conversion to $CO_2$, respectively. The $^{14}CH_4$ technique quantifies the amount of $^{14}C$ – initially supplied as methane – that is fully oxidized and persists as soluble species ($HCO_3^-$) or acid-labile precipitation products ($CaCO_3$). The $CH_3D$ protocol, on the other hand, reports the extent to which methane-derived hydrogen atoms are detected in water." Thus, the $^{14}CH_4$ method tracks carbon and measures fully oxidized inorganic carbon products, while the $CH_3D$ approach tracks hydrogen and measures methane-derived water-exchangeable hydrogen atoms, which may or may not correspond to fully oxidized methane.

**30.** L313: Change to: Because methane is an inert molecule abiotic exchange between methane and water protons is not expected, and indeed the D/H ratios in controls remained stable. Use term significantly/ significant should be reserved for statistical analyses. Not here. Please show those results and any measurement somewhere – the reference to Fig.1 is inappropriate- I don't see isotope values there.

We appreciate the reviewer's requests, and have altered this section accordingly. It now reads as follows (lines 384-388): "Because methane is an inert molecule, abiotic exchange between methane- and water-associated hydrogen atoms is not expected. Indeed, D/H ratios in killed control experiments remained stable (e.g., exhibiting a value of $1.40 \times 10^{-4}$ +/- $3.1 \times 10^{-8}$ SE at $T_{0d}$ and $1.40 \times 10^{-4}$ +/- $2.9 \times 10^{-8}$ SE at $T_{140d}$ during experimentation with *M. trichosporium*, data incorporated into Fig. 1a)."

**31.** L323: If you refer to backfluxes such as in Holler et al., you should allow the backfluxes throughout the model (figure 4). And based on this and further studies (i.e. Yoshinaga et al. ,2014) relative backfluxes in AOM should increase with lower / no methane

We thank the reviewer for pointing out this inconsistency in the anaerobic methanotrophic pathway (Figure 3). The backfluxes have now been incorporated with additional arrows and the caption has been adjusted accordingly. The relevant portion of the caption now reads as follows (lines 804-807): "Shorter "backflux" arrows reflect the observation that all enzymes (Thauer,

2008) and the entire pathway (Holler et al., 2011) have been shown to be reversible. For figure simplicity, isotopically distinct backflux products and cofactor involvement in backflux reactions are not shown."

The sulfate- and/or methane-dependent nature of relative backfluxes is also a useful observation; however, such a detailed distinction is incidental to the larger message of this sentence, and the Yoshinaga reference was not added here (although it is cited later in the discussion…).

**32.** 3.3.1. The H:C Tracer Ratio in Anaerobic Methanotrophy (Sorry, again this title is less than sloppy). Relation methane turnover rates derived from D and C labeling ? or something similar

Please see our response to comments #15 and #28 above.

**33.** L327-360 I cannot follow this discussion, especially from L332, I have a vague idea what the author wants to tell. The reversibility of specific steps, in particular of the activation step will lead to higher apparent rates in D compared to C labeling.

This portion of the discussion attempts to account for the values of the D:$^{14}$C tracer ratios observed in our anaerobic methanotrophy experiments. A full accounting of hydrogen and carbon atom flow through methanotrophic pathways is beyond the scope of this manuscript, but studying D:$^{14}$C tracer ratios offers useful insight that can inform future isotope probing studies. We have adjusted several portions of section 3.3.1 to express our analysis and its motivation in a more straightforward way.

**34.** 379: Quantifying anaerobic methane oxidation at different methane pressures

See response to comment #15 for section title adjustments.

**35.** 3.5 I would put the monodeuterated methane experimental investigations further up to the results. The interpretation of these experiments is really not well judged.

Section 3.5 compiles the lessons learned from the preceding results and discussion into three ready-for-use recommendations for researchers to gain insight and propose additional investigations through the use of the monodeuterated methane method. In this context, its placement at the end of the Discussion section is appropriate, and it was not moved.

We agree that the results are difficult to interpret and that a full accounting of hydrogen and carbon atoms in our methanotrophic experiments is underdetermined. However, such detailed accounting of the physiological controls on hydrogen exchange by aerobic and anaerobic methanotrophs is beyond the scope of this work and, we believe, is not critical for an initial description of the monodeuterated methane method provided in this manuscript. This treatment is akin to that taken with the previously published tritiated methane method (Valentine et al., 2001; Bussmann et al., 2015). The approach has several logistical advantages, is precise, reliable, easy to implement in the field, and offers an additional stable isotope tool with which to probe methane-associated metabolisms.

**36.** L421: Why do you think one would have less isotope fractionation effects in D than in C?

The phrase in question is intended to refer to the fractionation of hydrogen, comparing deuterated methane with tritiated methane. We would expect a smaller isotope effect in this context because the proportional mass difference between H and $^2$H is less than that between H and $^3$H. We realize our language was imprecise, and the phrase now reads as follows (lines 505-506): "…may be less susceptible to hydrogen-associated isotope fractionation effects (relative to $^3$H)."

**37.** L422: The precision to meet a chemically measured value was unfortunately not tested at all – so the results might be highly reproducible but we don't know yet what they show.

Please see our response to comment #18 above.

**38.** L423ff: Please state what both methods achieve: In aerobic methanotrophy D labeling might be more realistic to reproduce total complete oxidation rates of methane than the methane carbon labeling, because methane carbon is partly assimilated. In AOM C-labeling should be better to track methane oxidation, as a backflux would be small. D-labeling of methane likely overestimates methane oxidation rates, as rate – determining backflux can occur at all steps.

As suggested in this comment, the full accounting of methane-derived hydrogen and carbon atoms in aerobic and anaerobic methane oxidation is underdetermined, and providing a full accounting of their fates is beyond the scope of this study. However, analysis on these issues is provided in section 3.3; we believe it would be unnecessarily redundant to repeat such analysis in section 3.5.

**39.** 4. Conclusion. You might have been precise with your measurement but accuracy or even trueness of your measurement was not tested

The monodeuterated methane approach to probing methanotrophic metabolisms is predicated on the tracking of hydrogen atoms into the aqueous phase. This approach is conceptually similar to the tritiated methane method published by Valentine et al., (2001), but instead offers a non-radioactive option for tracking the fate of methane associated hydrogen. Because hydrogen atoms equilibrate with the water phase and are incorporated into biomass in poorly constrained ways, we would not expect – nor do we state in the manuscript – that our approach would be a true method of quantifying full methane oxidation from methane to carbon dioxide. The salient point for the conclusions presented in this manuscript is that the measured D/H ratios are consistently proportional to the well-established $^{14}$CH$_4$-based method for a given experimental system. Thus, aqueous isotopic ratios can be converted to a "full methane oxidation" value.

We believe that these key points are sufficiently expressed in the first paragraph of the conclusion, which reads in part as follows (lines 561-570): "The monodeuterated methane technique presented here represents a novel approach to methane oxidation rate measurements, notable for its logistical and analytical ease (particularly in ship-board applications), as well as the added dimension provided by H-based, rather than C-based, information. We have demonstrated that the D/H ratio is a reliable proxy for methane oxidation activity: in several

applications, the value is directly proportional to methane oxidation rates as measured in absolute terms by the well-established $^{14}CH_4$ method. The value of the proportionality constant differs based on the experimental system, likely dictated by environmental variables and the relative proportions of aerobic and anaerobic methanotrophic metabolisms, though additional experiments to determine the nature of the putative mixing line are needed."

**Reviewer 2**

The manuscript describes a novel technical approach to measure biological methane oxidation and track H-atoms through methanotrophic metabolisms. In part one of the manuscript, the authors focus on a comparison between the new CH3D method with the established 14CH4 method. For comparison of the results of the two methods a scaling was determined. To evaluate their method the authors performed measurements on methanotrophic culures and environmental sediment samples. Part two describes how the CH3D approach can be used to track H-atoms in anaerobic an aerobic methanotrophic pathways. Part three deals with a pressure experiment in which sediment samples were incubated at 9 MPa and 0.1 MPa to discuss the effect of pressure on methane turnover. The title of the manuscript describes the work adequately and the manuscript is well structure and written. Based on the few comments below, I suggest minor revision before the manuscript will be published.

Specific comments

Abstract

The abstract is well written and nicely reflects the outcome of the work.

**40.** Line 10. The poor precision of the established methods to determine a methane oxidation rate is mentioned (e.g. 14CH4). That is definitely true and an increase of precision is desirable. However, if precision is one of the major points that should be improved, the authors should tell the reader something about how (and in which range) the new method affect the precision of the measurements. That should be integrated in the abstract with a comparison in numbers of the precisions "14CH4 against CH3D" – derived from their comparative studies.

We agree that the enhanced precision of the $CH_3D$ technique compared with the $^{14}CH_4$ method is an important selling point; to support this contention, we have now incorporated calculations of improved precision into the manuscript. Lines 506-512 now read as follows: "Our results also suggest that that the monodeuterated methane technique appears to be a more precise method based on standard error calculations (Figs. 1, 2). Direct comparisons of environmental incubations are complicated by the microheterogeniety of seep settings (Barry et al., 1996; Lloyd et al., 2010), as well as the fact that different aliquots of the same initial material were used in our experiments. However, analysis of culture-based experiments reveals that standard errors from $R_{CH3D}$ values were 20% those derived from $^{14}CH_4$-based values, making the monodeuterated method five times more precise."

To reflect this finding in the abstract, lines 16-20 now read as follows: "We provide direct comparisons between the $CH_3D$ procedure and the well-established $^{14}CH_4$ radiotracer approach for several methanotrophic systems, including type I and type II aerobic methanotroph cultures – for which the new approach is five times more precise – and methane seep sediment and carbonate rocks under anoxic and oxic incubation conditions."

**41.** Line 14. The description of the central analytical device is rather weak "…using a water isotope analyzer". Since this is the new basic tool that allows this new approach, a subclause

about how the system works (Off-axis ICOS technology) should be already integrated in the abstract.

We appreciate the desire for additional methodological specificity in the abstract, and this suggestion has been incorporated into the text. Lines 10-13 now read as follows: "Here we present a new approach for measuring rates of microbial methane metabolism, using monodeuterated methane ($CH_3D$) as a metabolic substrate and quantifying the change in the aqueous D/H ratio over time using off-axis integrated cavity output spectroscopy." More information on the underlying technology and its previous scientific utility is provided on lines 245-249.

**42.** Line 21ff. The pressure experiment is very nice approach that shows how important incubations under in situ conditions are to determine real methane turnover rates. However, the story get lost in the abstract (just 2 lines) and appears a bit out of context (see comments below to chapter 3.4).

The pressure experiment provides an opportunity to demonstrate the $CH_3D$ approach in a scientifically valuable context. To make this case more strongly, this passage has been expanded to two sentences (lines 20-24) and reads as follows: "We also employ this method in a non-traditional experimental set-up, investigating the role of pressure on methane oxidation rates in anoxic seep sediment. Results revealed an 80% increase in methanotrophic rate at the equivalent of ~900 m water depth (40 MPa), revealing an important environmental parameter for methane metabolism and exhibiting the flexibility of the newly described method." Additional detail justifying the inclusion of this experiment is provided in section 3.4.

**43.** Line 26ff. Point 2 is difficult to understand without reading the manuscript. The phrase "scaling factor" is difficult to understand without a deeper context. I would suggest reformulating and extending the sentence.

In order to more fully describe the three proposed use cases for the method presented in this manuscript, this section of the abstract has been restructured, and point 2 has been expanded as follows (lines 27-31): "Second, by empirically linking the $CH_3D$ procedure with the well-established [14]C-radiocarbon approach, an absolute scaling factor can be determined for new systems of interest. This "ground-truthing" strategy enables "$CH_3D$ only" experiments to yield rates of full methane oxidation; we demonstrate this principle in the context of several methanotrophic systems."

Introduction

The introduction is well structured and gives appropriate background information.

**44.** Line 46-55. AOM is introduced but no background information about aerobic methane oxidation (MOB types) are delivered. This should be completed, because the lab test presented in the manuscript are not only focused on AOM.

We thank the reviewer for pointing out this gap in our introductory section. The phylogenetic and metabolic background of aerobic methanotrophy is now summarized in lines 63-71, which read as follows: "Methane is oxidized aerobically by members of the classes *Gammaproteobacteria* (Type I) and *Alphaproteobacteria* (Type II); verrucomicrobial representatives were more recently found to perform aerobic methanotrophy under extremely acidic conditions (Dunfield et al., 2007; Op den Camp et al., 2009). Methane monooxygenase converts methane to methanol, which is further oxidized to formaldehyde; assimilatory pathways branching at this point can incorporate carbon into central metabolism through the ribulose monophosphate (RuMP) cycle (Type I methanotrophs) or the serine cycle (Type II). Remaining formaldehyde can undergo two additional oxidation reactions, being converted first to formate and ultimately to carbon dioxide (Hakemian and Rosenzweig, 2007)."

**45.** Line 57. "biogeochemical curiosity", please rephrase or explain what you mean in more detail.

To clarify our intended message that methanotrophic metabolisms are biochemically underdetermined – an assertion that drives the desire to track elemental flows through pathways to better understand how metabolites are processed – the phrase "a biochemical curiosity" has been replaced by "a poorly understood biochemical process."

**46.** Line 71. References. I would also add a more recent paper, because the methods used changed a bit (e.g. I. Bussmann et al., Assessment of the radio 3H-CH4 tracer technique to measure aerobic methane oxidation in the water column, L&O Methods, 2015)

This reference has been added; we thank the reviewer for pointing us toward this recent paper.

**47.** Line 84ff.Why did the authors decided to test their new approach against 14CH4 and not also against tritium labeled methane with its improved specific activity that allows incubations under more realistic methane concentrations.

Comparing the monodeuterated methane approach to the $^3$H-CH$_4$ tracer method would indeed be a valuable point of reference. However, an exhaustive comparison using the full range of sample types discussed in this manuscript (cultures and environmental sediments and carbonates under oxic and anoxic conditions) was beyond the scope of our study. Logistical considerations, including a local collaborator with $^{14}$CH$_4$ expertise and more common community use of $^{14}$CH$_4$ approaches (see response to comment #18 above) also played a role. In addition, because the tritiated methane approach has been "ground-truthed" to the $^{14}$CH$_4$ procedure (Pack et al., 2011; Mau et al., 2013), links to our monodeuterated methane method can be made via the transitive property. We have added a passage in the discussion (lines 533-536) promoting more direct comparative study: "We also encourage side-by-side comparisons with other rate measurement approaches, including $^3$H-CH$_4$ radiotracer and methane concentration assessments, to develop additional pairwise conversion factors and better constrain carbon and hydrogen metabolism in methane-based biological reactions."

**48.** Line 85. Please explain in more detail what you mean with "partial versus complete methane oxidation".

By following hydrogen atoms into the aqueous phase, the monodeuterated methane approach yields signal throughout the methane oxidation pathway (see Fig. 3), even if carbon is not fully oxidized. On the other hand, the $^{14}CH_4$ method quantifies full oxidation by measuring dissolved inorganic carbon species. While this idea is expanded upon later in the manuscript, we have clarified the message here; this sentence now reads as follows (lines 104-107): "The resulting ratios, when viewed in the context of partial (methane-associated hydrogen exchange) versus complete methane oxidation (methane oxidation to $CO_2$), represent a new tool with which to examine the reversibility and catabolic / anabolic partitioning of methanotrophic metabolisms."

**49.** Line 91ff. This sentence is redundant and could be deleted.

This suggestion has been incorporated into the current version of the manuscript; we agree that the advantages of the monodeuterated methane approach are adequately summarized in the preceding paragraph, and explained at length in the discussion and conclusion sections.

**50.** Line 82ff. I cannot find any hint to the pressure experiment in this outlook. Such an outlook should cover the main aspects discussed in the following text.

We thank the reviewer for pointing out this oversight. The following sentence has now been added to the final paragraph of the introduction (lines 107-110): "As a proof of concept, we apply the monodeuterated methane approach to pressurized methane seep sediment incubations in order to test the role of an understudied environmental variable in methanotrophic rates under non-traditional empirical conditions."

Methods

The chapter is well structured and explains the different methods in an appropriate way.

**51.** Line 100ff. As mentioned above, also here the pressure experiment is a bit out of context. Why did the authors decide to perform these experiments without a comparison with 14CH4 rate measurements? See also comments below for chapter 3.4.

The use of the $CH_3D$ approach in the pressure-based experiments showed the method's transferability to new experimental set-ups. In particular, the approach does not require specialized equipment (such as suspended scintillation vials) that makes the $^{14}CH_4$ option cumbersome. This advantage of monodeuterated methane is mentioned briefly here and explained in further detail in section 3.4. The sentence in question in section 2.1 now reads as follows (lines 127-131): "In addition, the monodeuterated methane protocol was employed in a pressure-based experiment to demonstrate the technique's adaptability to distinct empirical set-ups and examine the relative effect of heightened, environmentally relevant pressure on methane consumption rates in anoxic seep sediment samples."

**52.** Line 130. Is the information about "unique four-digit serial number" needed? I think the sentence can be deleted.

This sentence has been removed in order to improve the flow of this section. However, we retain the four-digit serial numbers and note their presence in the caption of Table 1, part of which now reads as follows (lines 776-778): "The three-part codes for samples derived from environmental material refer to active (A) or low-activity (L) sediments (Sed) or carbonates (Carb), along with a sample-specific four-digit serial number." Keeping these serial numbers is important; other aspects of these samples or those collected nearby may be subjected to additional analyses in multiple labs. Being able to link future data to the rate measurements provided here may bolster the interpretive power of such studies.

**53.** Line 131. Maybe insert: "The "active" designation in our sample description (e.g. Figure 2) refers…"

This suggestion has been incorporated into the text; the passage now reads as follows (line 163): "The "active" designation in our sample descriptions refers to sites…"

**54.** Line 141. That is difficult for me to follow. A few sentences before the authors say that carbonates are formed during "active" periods of seepage (line 137) and now they say that carbonates (L. Carb) can also exist under "low-activity". Please explain the difference between L. Carb and A. Carb in more detail.

In order to streamline this paragraph and avoid confusion, we have omitted the following sentence: "The presence of carbonate pavements, coupled to depleted $\delta^{13}C_{carbonate}$ values suggest that they formed during "active" periods of seepage, consistent with previous descriptions (Naehr et al., 2007; Peckmann and Thiel, 2004)." This sentence suggested that methane-derived carbonates are believed to form primarily during periods of "active" seepage rather than produced *in situ* at "low-activity", off seep sites. While the carbonates are produced in active seep sites, the carbonates themselves persist long after the decrease or termination of methane advection and thus can be sampled from a range of habitats, including active, low-activity, and inactive seep sites. We hope that this more focused paragraph (and the papers referenced therein) is more helpful to readers.

**55.** Line 147ff. I would suggest to move the entire paragraph to line 130. First describe how the samples were taken and then how the samples were named (paragraph 130ff).

We appreciate the reviewer's request for a more logically structured methods section, and have adjusted these paragraphs accordingly. The first sentence from the (previously) line 147 paragraph now precedes the (previously) line 130 paragraph. This arrangement first explains how samples were taken (lines 160-163), then describes the sample sites (lines 163-175), and finally shows how samples from these sites were processed, first shipboard (lines 176-179), and then in the lab (lines 179 onward).

**56.** Line 153. The samples were stored in Ar-flushed bags. Does this influence the methane concentration in the sample and also maybe the activity of the microorganisms? How long were the sample stored?

The purpose of argon flushing is to maintain an anoxic and dormant state prior to proper experimental set-up, such that the microbial community is not substantially altered or inhibited (by product accumulation in a closed system, for example) before the study can commence. To the best of our knowledge, a thorough examination of this process's effect on community structure and subsequent activity has not been done. Nonetheless, this step is a standard part of maintaining sufficiently anoxic conditions during shipboard collection to enable the re-activation of AOM back in the lab with the addition of methane. Samples were stored in this fashion for several months before experimental set-up, now specified in the text (lines 177-179).

**57.** Line 155. The samples were maintained under 2x10^5 Pa CH4 headspace for one month. Why one month? And how does that fit to in situ conditions (methane concentration)? And if there are differences can we expect that it also influences the activity of microorganisms in the experiment? A comment on that should at least be given in the discussion somewhere.

Because the argon flushing process generates a state of dormancy, some time under methane-replete conditions is required to re-start methanotrophic biological processes. We have found that ~one month of start-up time is sufficient to resuscitate activity in these slow-growing organisms (average doubling time of 3 months) in methane seep sediments, without substantially altering the microbial community (based on 16S rRNA Illumina TAG sequencing). The $2x10^5$ Pa $CH_4$ headspace leads to a dissolved concentration of 3.7 mM. This information is now provided in lines 182-185, which read as follows: "Samples were maintained under a $2x10^5$ Pa $CH_4$ headspace for one month in order to resuscitate activity; the corresponding dissolved concentration (3.7 mM) is consistent with environmental methane concentrations at Hydrate Ridge."

**58.** Line 166. Which gas was injected – CH4? And why does it end in a desirable headspace composition?

The injected gas varied depending on the nature of the experiment – anoxic or oxic – being set up. This section has been streamlined (the "desirable" description has been removed to avoid confusion), and now reads as follows (194-198): "All bottles were sealed with butyl stoppers; following several minutes of flushing with $N_2$ (g), the headspace was replaced with methane, and an additional 30 mL of gas, whose composition varied depending on the experiment, was injected into the 30 mL headspace to generate an absolute pressure of approximately $2x10^5$ Pa. Anoxic incubation headspace was 100% methane; oxic incubation headspace was 30 mL methane, 20 mL $N_2$, and 10 mL $O_2$."

**59.** Line 168. What was the reason to choose this specific gas composition? Does it reflect environmental conditions?

This mixed gas composition was chosen in attempt to maintain aerobic methanotrophic growth (it had been shown to do so for the microoxic marine gammaproteobacterial methanotroph *M. sedimenti* (Tavormina, pers. comm.)), while simultaneously maintaining anoxic niches for anaerobic methanotrophs. This latter aspect accounts for the <50% $O_2$ headspace concentrations, in contrast to what was used for the aerobic cultures (50:50 methane:oxygen). The intermediate

D:$^{14}$C tracer ratios that these oxic environmental incubations produced suggests that both types of methanotrophs were indeed active (see Table 2, discussion section).

**60.** Line 180. What are "mylar" bags. Are they gas tight? Maybe a short comment on that in the text.

Mylar bags are indeed gas tight; this important point has been added at the first mention of the material, on lines 178-179.

**61.** Line 180. Actually, I could not find Table S2 in my documents and therefore cannot comment on that.

Please see http://www.biogeosciences-discuss.net/bg-2016-202/bg-2016-202-supplement.pdf for Table S2 and all other supplementary material. (Updated versions of all figures and tables are included in our response to reviewer comments.)

**62.** Line 188. Why did the authors choose 9.0 MPa. Where does the sample come from (water depth, temperature). Some more comments on the sample are needed.

The parameters chosen for the pressurized experiment (~900 meters below sea level depth equivalent, 4 °C) correspond with the environmental conditions from which the sediment sample was collected (850 mbsl, 4 °C). These details have now been added to the sample description on lines 208-209.

**63.** Line 189. The authors tested visually the bags for leaks. I think a better method to test for leaks would be the analyses of CH4 (or CH3D) in the water of the pressure vessel at the start point and end point of the experiment. That would also deliver information about diffusion of methane through the bag into the surrounding water. Can diffusion be excluded?

We appreciate the reviewer's concern about the mylar bags' integrity during this pressure-based experiment, and the proposed tests would provide additional confidence that no methane leaks occurred. However, we are confident that substantial leakage or diffusion did not occur for two reasons. First, after submerging the sealed mylar bags in the oil-filled pressure chamber, no oil was observed inside the bags following the experiment. Second, upon removal from the pressure chamber, the bags quickly re-inflated to their full pre-pressurization sizes. The speed of this expansion suggests that there was no gaseous diffusion out of the bag during the experiment, as such rapid diffusive re-inflation does not occur through mylar at atmospheric pressures. Furthermore, the gas-tight nature of mylar bags has been well established through multiple studies (e.g., Cohn et al., 1963; Caspersen et al., 2013)

**64.** Line 194. How was the volume of the water sample replaced in the culture?

The replacement of liquid volume during sampling is an important procedural step – particularly with the smaller-volume cultures – and we thank the reviewer for pointing out its omission. A sentence describing our volume replacement approach has been added (lines 240-242): "A

constant volume was maintained by adding 1 mL of sterile media immediately after sampling; this media was previously equilibrated with gaseous headspace specific to each experiment."

**65.** Line 196. The only information about the main analytical device is the name of the model and the company. Since this tool represents something that is really new in the context of methane rate measurements, I would like to have some more details about the main analytical principle of the system (Off-axis ICOS technology). Can the authors deliver any additional references to the system (other studies)?

We thank the reviewer for pointing out this oversight, which has been rectified with the addition of one sentence to provide additional technical detail on the instrument, and another providing examples of previously published research applications. This section (lines 243-249) now reads as follows: "A LGR DLT-100 liquid water isotope analyzer (LWIA, Los Gatos Research, Mountain View, CA) was used to determine the D/H ratio of each sample. The LWIA uses off-axis integrated-cavity output spectroscopy to measure isotopically specific absorption patterns and determine simultaneous D/H and $^{18}O/^{16}O$ ratios with high precision and robust mechanics (Lis et al., 2008). Such instruments have been used for a range of studies, including hydrological analysis (Robson and Webb, 2016), mine waste management (Huang et al., 2015), and microbial metabolism (Dawson et al., 2015)."

**66.** Line 205ff. What does it mean "sub-optimal". Is a statistical test behind that?

The temperature and pressure alarms are automatic responses to conditions that limit the water analyzer's optimal performance. The temperature alarm is raised when the temperature of the water vapor inside the measurement cell changes at a rate exceeding 0.3 °C per hour; the pressure alarm appears when the pressure inside the measurement cell rises during a measurement. This sentence has been modified to include these details, and now reads as follows (lines 257-260): "Data was removed if instrumental temperature or pressure parameters were observed to fall outside of optimal instrument specifications (0.76% of all analyses), corresponding to an internal temperature change of more than 0.3 °C per hour or rising pressure within the measurement cell during the analysis."

**67.** Line 211. I am not sure if I understood this part correctly. Is the assumption of a linear scaling factor only based on two standards? Is the LGR system linear over the measurement range? Was this tested?

Yes, this is correct, and is now specified in additional detail in lines 261-263: "To calculate methane consumption rates, D/H ratios from the LWIA were first normalized to the Vienna standard mean ocean water (VSMOW) scale using a two-point calibration from the water standards and a linear interpolation (e.g., Dawson et al., 2015)." The two-point calibration approach across similar absolute δD ranges has been previously published by Dawson et al., 2015, and subsequent tests using four standards between -123.7‰ and -9.2‰ has shown a nearly perfect linear relationship with an $R^2$ value of 0.9999342.

Results and Discussion

**68.** Figure 1. It would be easier for the reader to follow the discussion, if Fig. 1a and 1b would be tilted with the name of the two MOB. I would also add a legend into the figures to explain the different symbols. The axis labels and the numbering on the axis do not look very accurate: the positions of the axis labels at the y-axis is not centered in both figures; 0 on the x-axis cuts the y-axis.

We appreciate the reviewer's suggestions here, and the comments on Figure 1 have all been incorporated into the manuscript. The figure caption has been adjusted accordingly to avoid repetition with the legend.

**69.** Line 271. "Using data points..." please list the data points used to derive the ratio in the text. Not only time also the methane oxidation rates.

This information has now been incorporated into the text. Lines 336-346 now read as follows: "Scaling factors were calculated for both exponential growth and stationary phase, using data points from both $CH_3D$ and $^{14}CH_4$ experiments. For example, the *M. trichosporium* rate value calculated from $CH_3D$ experimental treatment point (47.5 hours, 4.16 x $10^4$ nmol methane consumed) was compared with the rate determined from $^{14}CH_4$ experimental treatment point (47.5 hours, 2.77 x $10^4$ nmol methane consumed), yielding a scaling factor of 1.5 for exponential phase growth. Similarly, data from (140 h, 5.27 x $10^4$ nmol, $CH_3D$) and (166.5 h, 4.24 x $10^4$ nmol, $^{14}CH_4$) were used for *M. trichosporium*'s stationary phase scaling factor. Equivalent values were determined for *M. sedimenti* using the following data points: (140 h, 7.07 x $10^3$ nmol, $CH_3D$) and (102 h, 3.43 x $10^3$ nmol, $^{14}CH_4$) for the exponential growth phase, and (476 h, 7.53 x $10^3$ nmol, $CH_3D$) and (432 h, 4.30 x $10^3$ nmol, $^{14}CH_4$) for stationary phase."

**70.** Line 281. To which table or figure do these numbers (e.g. #1b) belong to?

Given the data points specified between lines 336-346, the sentence to which this comment refers is no longer necessary and has been deleted from the current version of the manuscript.

**71.** Line 284. Does the 14CH4 method yield the "full-oxidation methanotrophy"? I think the correction of the CH3H oxidation rates using the H:C tracer ratio can just deliver oxidation rates, which can be better compared with the rates obtained with 14CH4 rate measurements.

The reviewer's distinction between "oxidation rates" and "full-oxidation methanotrophy" is not quite clear to us, so we have clarified what we mean by "full-oxidation methanotrophy." The relevant phrase now reads as follows (lines 355-357): "By dividing rates derived from D/H values by 1.5, a reliable estimate of full-oxidation methanotrophy – that is, the complete biological oxidation of methane to carbon dioxide – can be assessed." This distinction is needed because the term "methanotrophy" does not distinguish between the complete oxidation of methane to carbon dioxide, and methane activation, whereby methane-associated hydrogen atoms may be enzymatically exchanged with water at multiple steps along the methane-oxidation pathway but the final, fully oxidized $CO_2$, may not result (for example, due to back flux reactions described in Holler et al., 2011).

**72.** Line 291. Please specify what you mean with "second time point"? IN which table or figure can I find the numbers 4d or 8d?

As mentioned in section 2.1.2, "measurements were taken for both D/H and $^{14}$C analysis at 1.9 and 4 days for oxic incubations, and 3 and 8 days for anoxic incubations;" it is the latter time points in each case that are shown in Fig. 2. To avoid confusion, the reference to multiple time points has been removed from section 3.2, and the relevant sentence now reads as follows (lines 362-364): "Values were calculated from data collected after 4 days of incubation for oxic samples and after 8 days of incubation for anoxic samples."

**73.** Line 371. Is any data available from the experiments to determine the cell density?

As referenced in section 2.1.1, aerobic methanotroph cultures exhibited substantial increases in optical density, and further relevant details of *M. trichosporium* and *M. sedimenti* growth dynamics can be found in the literature (e.g., Whittenbury et al., 1970 and Tavormina et al., 2015, respectively). Cell density data is not presented in this manuscript. However, we agree with the reviewer's intimation that by combining aqueous D/H data with biomass assessments, further resolution on the fate of methane's hydrogen atoms could be attained. Such an investigation would be bolstered by lipid or compound-specific δD measurements of biomass throughout a growth curve, but this work is beyond the scope of our report.

**74.** Chapter 3.4 As mentioned before, I have the feeling that this part of the manuscript is a bit out of the main focus. It is for sure an interesting approach but if this approach would be extended (more samples, different simulations (e.g. pressure),...), it could stay for itself. My main question are: What is the goal of these pressure studies? To show that pressure influences methane turnover? What is the advantage of the CH3D method compared with the 14CH4 method for these kind of pressure experiments?

I am sure that the influence of in situ pressure is more important for the outcome of the experiment than the use of the new CH3D rat measurement approach (e.g. higher precision?). I think it must be explained in more detailed why exactly such an experiment can help to evaluate the now CH3D approach (without having data from a parallel 14CH4 approach).

As briefly mentioned in response to previous comments, the pressure study provided an opportunity to demonstrate the ease of use and scientific value of the monodeuterated method in a challenging experimental set up. Subjecting seep sediment samples to elevated pressures involved placing them in mylar bags, which form an impermeable but flexible boundary. This flexibility is essential for transmitting pressure in an oil-filled chamber, but it makes traditional methanotrophic measurements via $^{14}$CH$_4$ very difficult and adds to the challenge of preventing radioactive contamination. The radiolabel method involves a number of standardized steps whose adoption in a mylar bag context would require substantial method development. For example, processing of post-incubation headspace is optimized for stoppered bottles; accessing the gas phase from a mylar bag in a quantitative fashion is not straightforward. The quantification of radiolabeled dissolved inorganic carbon produced by methanotrophic activity requires that all incubation material be transferred to an Erlenmeyer flask in which a plastic vial

filled with scintillation fluid is suspended from a rubber stopper. Sediment grains are commonly trapped in the seals of mylar bags, making this quantitative transfer challenging.

Portions of the introduction, methods, and discussion sections have been augmented to describe the unique advantages of performing these experiments with the $CH_3D$ technique. Lines 107-110 in the introduction now offer a stronger justification for the pressure-based experiments as a useful testing ground for the $CH_3D$ approach. "As a proof of concept, we apply the monodeuterated methane approach to pressurized methane seep sediment incubations in order to test the role of an understudied environmental variable in methanotrophic rates under non-traditional empirical conditions."

Lines 218-226 in section 2.1.3 now read as follows: "The use of flexible mylar bags is essential for the application of external pressure, yet it presents obstacles for "traditional" methanotrophic rate measurement protocols such as the $^{14}CH_4$ method. In particular, the processing of post-incubation headspace is optimized for stoppered bottles, and accessing the gas phase from mylar bags in a quantitative fashion is challenging. Measurement of radiolabeled dissolved inorganic carbon requires that all incubation material be transferred to an Erlenmeyer flask equipped with a scintillation vial; sediment grains are commonly trapped in the seals of mylar bags, complicating this transfer. For these reasons, monodeuterated methane addition and subsequent aqueous measurement offered a useful tool for this challenging experimental set-up."

Section 3.4 has been modified to convey these points, and the first paragraph of this section (lines 461-471) now reads as follows: "To demonstrate the utility of the $CH_3D$ rate measurement approach in addressing experimentally relevant questions, particularly in nontraditional empirical contexts, we sought to evaluate the influence of *in situ* pressure on methanotrophic rates of Hydrate Ridge seep sediment microbial communities. Material collected for microbiological studies of AOM is frequently obtained from marine settings of various depths that are subjected to distinct and substantial pressure regimes (Ruff et al., 2015). Pressure is not always rigorously incorporated into microcosm experiments, though evidence suggests it can be an important determinant of methanotrophic rates (Bowles et al., 2011; Nauhaus et al., 2005; Zhang et al., 2010). In addition, some procedural aspects of the $^{14}CH_4$ protocol, including headspace sampling and full-volume transfer, are not established for use with mylar bags, making the monodeuterated methane approach an appealing alternative in this context."

**75.** Line 392. Isotopically labeled glycine and ammonium chloride was not mentioned before in the manuscript. Please give detailed information about this experiment already in the first part of the manuscript (e.g. paragraph 82ff). For what is good for? What is the goal of that labeling experiment?

We thank the reviewer for pointing out this omission, and a few modifications have been made. In section 2.1.3, a sentence explaining the different nitrogen sources has been added; lines 212-213 now read as follows: "500 µM glycine or 500 µM ammonium were added in order to evaluate relative rate differences associated with organic and inorganic sources of nitrogen." In the discussion, we offer a brief interpretation of the distinct effects of nitrogen sources (lines 479-482): "Incubation with 500 µM glycine rather than ammonium at high and low pressures resulted in small but consistent rate increases of 12% +/- 4.1% SE, potentially reflecting the

energetic and biosynthetic distinction between exogenous amino acids and unprocessed fixed nitrogen."

We did not highlight the addition of different nitrogen sources in the introduction, as we did not see this as a "top-level" question that needs to be previewed in the opening section. In order to minimize confusion for the readers, for the purposes of this manuscript, we now have modified Table S2 to mention the glycine and ammonium additives only as nitrogen sources.

**76.** Line 417. Please explain why the pressure experiment is a proof-of-concept. Pressure makes the difference in this experiment not the method that was used for methane oxidation rate measurements.

The pressure experiment was a demonstration of the monodeuterated methane approach in the sense that it generated primary data in support of a compelling scientific question. Please see response to comment #74 above for additional thoughts and modifications that have been made to the manuscript.

**77.** Line 437. One advantage of the CH3D method is that "it does not require the logistical, safety, and administrative hurdles associated with radiotracers such as 14CH4..." (line88ff). But to obtain absolute rates of full methane oxidation, parallel incubations with CH3D and 14CH4 must be performed. That means that we still have to take radiotracer on ships (together with the CH3D lable and analytical equipment) with all the administrative hurdles. That means no advantage for expeditions?

The reviewer's note that experimental systems previously untested with monodeuterated methane will require a "ground truthing" $CH_3D$ – $^{14}CH_4$ side-by-side experiment is correct. However, we believe the $CH_3D$ approach still offers logistical advantages for expeditions, particularly if an absolute quantitative rate of complete methane oxidation is not required shipboard (i.e., to measure relative rates) or in situations where methanotrophic activity has been previously established and a known D:$^{14}$C tracer ratio can be applied (e.g., methane seeps). Following the establishment of a satisfactory D:$^{14}$C tracer ratio, additional experiments could theoretically be performed with only $CH_3D$; this feature is mentioned in the discussion, lines 527-530: "Second, by performing side-by-side monodeuterated methane and radiocarbon tests, a sample-specific D:$^{14}$C tracer ratio can be determined, and absolute rates of full methane oxidation can then be inferred in subsequent experiments based exclusively on D/H ratios."

**78.** Figure 1 See comment above Colors are difficult (e.g. I cannot see brown on my printout). Would suggest to change the colors.

The colors used in this figure make use of a wide dynamic range of user-friendly (i.e., dark-hued) colors. Rather, data points may not be visible because other data series are plotted in front of them; minimizing circle sizes to reveal all data points would render them nearly invisible. We believe this situation is adequately explained in the caption to figure 1, which reads in part as follows (line 792): "Obscured data points exhibited values between -60 and 110 nmol for a) and 0 and 60 nmol for b)."

**79.** Figure 3 and 4 Please give the figure titles like "anaerobic methanotrophy pathway" and "aerobic methanotrophy pathway".

This suggestion has been incorporated into the current version of the manuscript. Figure 3 is titled "Reverse Methanogenesis" Pathway, an appellation that is justified in the main text's first reference to the figure, which now reads as follows (lines 402-404): "AOM is depicted in Fig. 3 via the reverse methanogenesis pathway, which is believed to be enacted by ANME based on genetic (Hallam et al., 2004) and proteomic (Marlow et al., 2016) data." Figure 4 is titled Aerobic Methanotrophy Pathway.

**80.** Figure 5 A legend (and also a title) in the figure would be helpful. Capture: That the data comes from the pressure experiment should be mentioned

These helpful suggestions have been incorporated into the current version of the manuscript. The figure itself clearly shows the pressure and stable isotope treatment of each sample, while sample numbers refer the reader to Table S2 for more details. The modified caption now reads as follows (lines 815-818): "Pressure experiment results showing water $\delta D$ values with standard error bars of seep sediment samples following 38-day incubations with $CH_3D$ at 9.0 MPa (brown bars, "b" samples) or 0.1 MPa (pink bars, "a" samples). Additional details on sample treatments can be found in Table S2."

**Reviewer 3**

The authors tested a potential new method to use monodeuterated methane (CH3D) as a metabolic substrate for methane oxidation rate measurements by quantifying the change in the aqueous D/H ratio over time. In the study, two methanotrophic cultures and several environmental samples were used to compare the radio of the novel CH3D method with the existing and well established 14C isotope method. The new approach is complementary to existing radio (14C)- and stable (13C) carbon isotopic methods.

Because it isn't a stand-alone technique; stressing the alleged advantage of a non-toxic, rapid, and easy-to-use method is obsolete.

We believe there are several distinct use cases and advantages to the method presented in this manuscript, and that the logistical benefits described are appropriate. Until our understanding of hydrogen atom fates in methanotrophy is improved, deriving an absolute rate value from $CH_3D$ does indeed require co-experimentation with radiolabel methods. However, once that calibration is performed, subsequent experiments on similar systems can be performed with only the $CH_3D$ approach. In addition, comparative rate measurements – subjecting the same inoculum to different parameters – does not require a calibration (since the D:$^{14}$C tracer ratio was shown to be consistent for a given inoculum type even at distinct absolute rates). These points are emphasized throughout, and particularly in the abstract and section 3.5.

The overall structure of the manuscript is clear, focusing on the different metabolic pathways, which were approved by the new method by using different biological samples. The authors give a nice overview of the potential of the presented method, although the argumentation in parts of the manuscript isn't easy to follow.

We appreciate the reviewer's concern, and after making the modifications detailed in the responses above, believe the manuscript's analysis and discussion are more streamlined. In particular, the improved description of the monodeuterated methane-based calculations and clearer justification for the pressure-based section should make the manuscript more accessible.

From my perspective, the principle of the method isn't sufficiently explained (see comments below as well as the statements of the other referees). Also the not informative subtitles need to be improved!

Please see the response to comment #15 above.

I suggest minor revision before the manuscript will be published. The abstract is smooth to read and well structured. The introduction gives a nice overview of the main topics of the publication and is well structured.

**81**. L36 the biogeochemical cycles in BOTH natural freshwater and marine environments is mentioned, but in L39 only the estimated methane emission in marine settings is given. Can you show values of produced methane in natural freshwater and rice fields/wetlands/permafrost as well?

We thank the reviewer for pointing out this oversight, and a sentence addressing methane production in freshwater wetlands has now been added (lines 47-49): "In freshwater wetlands, approximately 200 Tg of methane is generated per year, most of which is oxidized by hydroxyl radicals in the troposphere (Kirschke et al., 2013)."

**82**. L47 biochemical intricacies – what is meant with that phrase?

This phrase refers to the details of methanotrophic metabolisms that have intrigued researchers for decades – energetics, enzymatic mechanisms, cofactors, etc. To clarify this thought, "intricacies" has been changed to "details".

**83**. L48 -55 AOM is described. Because aerobic methane oxidation is a major part of the experimental setup and the discussion, aerobic methane oxidation should be introduced as well.

We agree with this suggestion offered by multiple reviewers; the newly added lines 63-71 now discuss the phylogenetic affiliations and biochemical strategies of aerobic methanotrophs.

**84**. L57 biochemical curiosity – what precisely is meant by this?

In response to other reviewer comments, this phrase has now been changed to "a poorly understood biochemical process." As with comment #82, we hope to convey the unknown aspects of methane-utilizing metabolisms. With its ability to trace hydrogen atoms, monodeuterated methane provides a compelling opportunity to better understand these reactions, an assertion that is substantiated in the manuscript's discussion.

**85**. L58 AOM rate measurements have traditionally been conducted using a handful of techniques. – I understand this sentence as introduction for the following methods, which not discriminate aerobic or anaerobic methane oxidation. It should be rephrased, see RC1.

In accordance with this comment and comment #6 above, this sentence now reads as follows (lines 75-76): "The oxidation of methane in environmental samples has traditionally been studied using a handful of techniques."

**86**. L61 Stable isotope 13CH4 tracers – 13CH4 isn't a tracer. Better: 13C-labelled methane

This change has been incorporated into the current version of the manuscript, and now reads as follows (line 78): "Methane labeled with $^{13}$C can be used…"

**87.** L70 and the procedural advantages of working with a water-phase product rather than gaseous products – this is one of the main advantages? I think RC1 makes a good point.

Please see our response to comment #11 above. The relevant passage now reads as follows (lines 87-90): "Labeling with tritiated methane was introduced for water column aerobic methane oxidation measurements due to its higher specific activity and the procedural advantages of

working with a water-phase product rather than gaseous products (Bussmann et al., 2015; Valentine et al., 2001)."

**88**. L107 and 109 where did you get the cultures from? Are these maintenance cultures or ordered as pure culture from a company?

To describe where the two aerobic methanotroph cultures came from, lines 123-126 (the first mention of both strains) now read as follows: "Both techniques were applied to a) aerobic methanotrophic cultures of *Methylosinus trichosporium OB3b* (kindly supplied by Marina Kalyuzhnaya and Mary Lidstrom) and *Methyloprofundus sedimenti* (isolated from a deep sea whale fall; Tavormina et al., 2015)…"

**89**. L118 how did you see that the exponential growth phase was reached? Which test did you use?

Exponential growth phase was measured by optical density at 600 nm wavelength for balch tube cultures grown in preparation for the $CH_3D$ experiments. As these cultures were grown and transferred several times, a predictable growth curve was established, providing confidence that our sampling time points captured exponential phase. To better reflect this protocol, lines 145-148 now read as follows: "Samples for D/H analysis were taken at seven time points throughout 140-hour (*M. trichosporium*) and 476-hour (*M. sedimenti*) experiments. Sampling points were most concentrated around anticipated exponential growth phases as determined by optical density (600 nm) profiles of earlier rounds of culture transfers."

**90**. L130 All samples received a unique four-digit serial number. – Unnecessary.

Please see the response to comment #52 above; this point has been removed from the main text.

**91**. L151 What is "compacted sediment"? And would it be interesting to know from what depth below seafloor (which layer of the push core) the sediment comes from? Or is it the whole push core sediment, which was transferred into a bag? But still, from what depth below seafloor comes the sediment?

To avoid confusion and convey the salient aspects of sample set-up, the "compacted" modifier has been removed. The depth horizon of sediment used in our experiments is indeed an important parameter; the relevant portion of the methods now reads as follows (lines 179-181): "In advance of experimental set-up, carbonate samples and homogenized sediment from the 0-12 cm push core horizon were prepared…" To provide this information for the pressure-based experiments, lines 210-212 now read as follows: "To set up the incubations, eight 100 mL mylar bags were prepared with the components shown in Table S2 using homogenized sediment from the 0-12 cm horizon."

**92**. L152 What's a mylar bag? There is a reference for the glass bottle (L158) but not for the bag. In general: check references for lab equipment.

This omission has been corrected; the first mention of mylar bags now reads as follows (Lines 177-179): "To prepare material for future experimentation, sediment and carbonate rocks were stored in anoxic, Ar-flushed, gas-tight mylar bags (Impak Corp., Los Angeles, USA) at 4 °C until use several months later."

**93**. L171 1.9 days? Better use hours, if it is necessary to mention the exact time point.

This suggestion has been incorporated into the current version of the manuscript; lines 200-202 now read as follows: "Measurements were taken for both D/H and $^{14}$C analysis at 46 and 96 hours for oxic incubations, and 72 and 192 hours for anoxic incubations."

**94**. L190 how did you do the leak check?

Based on the observation that the bags were still gas tight and that no oil from the surrounding media was found inside the bags following removal from the pressure chamber, we are confident that no leaks occurred during the incubation. Lines 233-236 now read as follows: "At the conclusion of the experiment, mylar bags were removed from the chamber, checked for leaks (none were observed, as the bags were still inflated, the seal was still gas-tight, and no hydraulic fluid was detected in the interior of the mylar bags) and sampled for D/H ratio measurement."

**95**. L198 the water isotope analyzer determine the D/H ration of the sample – can you explain more detailed how the analyzer works?

Based on this and other reviewer comments, additional detail has been added. In particular, lines 243-249 now read as follows: "A LGR DLT-100 liquid water isotope analyzer (LWIA, Los Gatos Research, Mountain View, CA) was used to determine the D/H ratio of each sample. The LWIA uses off-axis integrated-cavity output spectroscopy to measure isotopically specific absorption patterns and determine simultaneous D/H and $^{18}$O/$^{16}$O ratios with high precision and robust mechanics (Lis et al., 2008). Such instruments have been used for a range of studies, including hydrological analysis (Robson and Webb, 2016), mine waste management (Huang et al., 2015), and microbial metabolism (Dawson et al., 2015)."

**96**. I am agree with both referees: a formula would be helpful to understand the method and the principle of the general measurement procedure. Maybe it is possible to visualize both in a schematic diagram.

Based on multiple reviewer comments, section 2.2.1 has now been substantially expanded to provide an annotated formula. We are also developing a schematic diagram (not yet in the current, revised version of the manuscript) to graphically portray the steps involved in the procedure.

**97**. L281 tubes #1a, #1b, and #1c – from which tubes/samples are you talking about? I assume you mean your replicates, then either delete the specification of tube labels and just talk about replicates or expand table 1 and include that kind of extra information.

This suggestion has been incorporated into the current version of the manuscript; based on the newly detailed description of the monodeuterated methane-based rate calculation, specific tube numbers are no longer needed. Lines 353-354 now read as follows: "D:$^{14}$C tracer ratio values were calculated for aerobic methanotroph cultures using the data specified above and are shown in Table 2…"

**98**. Chapter 3.4. I am not sure, if the pressure experiment gives an additional value to the manuscript or rather create more confusion. The main goal is to present the method and to explain the method in a way that it's clear and easy to understand – that is sometimes a challenge especially in Chapter 3.3.

We appreciate the need to better integrate the pressure-based experiments into the broader aim of the manuscript, and in response to this and other reviewer comments, several sections have been bolstered to this effect. Please see the response to comment #74 above for additional details. We present this experiment as a direct illustration of the value in using non-radioactive tracers in situations where the conventional $^{14}CH_4$ assay may be logistically challenging.

**99**. L454 is, where you bring up by the first time in the whole article for what the abbreviation NMR stands for. Should be mentioned earlier in L256

In order to clarify the acronym used throughout the manuscript, we have now defined it at its first appearance, on line 203.

---

## Author Comment (AC4) · 18 Aug 2016

Please see our responses to all reviewer comments in the attached pdf entitled "Responses to Reviewer Comments" (those pertaining to reviewer 3's comments are found on pages 25-29).

Please also note the supplement to this comment: http://www.biogeosciences-discuss.net/bg-2016-202/bg-2016-202-AC4-supplement.pdf
* * *